# Mitochondrial DNA is a target of HBV integration

Domenico Giosa[1,2,8], Daniele Lombardo[1,2,8], Cristina Musolino[2,3], Valeria Chines[1,2], Giuseppina Raffa[1,7], Francesca Casuscelli di Tocco[1,2], Deborah D'Aliberti[1,2], Giuseppe Caminiti[2], Carlo Saitta[1], Angela Alibrandi[4], Riccardo Aiese Cigliano[5], Orazio Romeo[6], Giuseppe Navarra[3], Giovanni Raimondo[1] & Teresa Pollicino [1,2 ✉]

Hepatitis B virus (HBV) may integrate into the genome of infected cells and contribute to hepatocarcinogenesis. However, the role of HBV integration in hepatocellular carcinoma (HCC) development remains unclear. In this study, we apply a high-throughput HBV integration sequencing approach that allows sensitive identification of HBV integration sites and enumeration of integration clones. We identify 3339 HBV integration sites in paired tumour and non-tumour tissue samples from 7 patients with HCC. We detect 2107 clonally expanded integrations (1817 in tumour and 290 in non-tumour tissues), and a significant enrichment of clonal HBV integrations in mitochondrial DNA (mtDNA) preferentially occurring in the oxidative phosphorylation genes (OXPHOS) and *D-loop* region. We also find that HBV RNA sequences are imported into the mitochondria of hepatoma cells with the involvement of polynucleotide phosphorylase (PNPASE), and that HBV RNA might have a role in the process of HBV integration into mtDNA. Our results suggest a potential mechanism by which HBV integration may contribute to HCC development.

[1] Department of Clinical and Experimental Medicine, University Hospital of Messina, Messina, Italy. [2] Laboratory of Molecular Hepatology, University Hospital of Messina, Messina, Italy. [3] Department of Human Pathology, University Hospital of Messina, Messina, Italy. [4] Department of Economics, University of Messina, Messina, Italy. [5] Sequentia Biotech Sl, Barcelona, Spain. [6] Department of ChiBioFarAm, University of Messina, Messina, Italy. [7] Present address: Laboratory of Molecular Hepatology, University Hospital of Messina, Messina, Italy. [8] These authors contributed equally: Domenico Giosa, Daniele Lombardo. ✉email: tpollicino@unime.it

Chronic hepatitis B virus (HBV) infection is a major risk factor for hepatocellular carcinoma (HCC) development. Compared to healthy individuals, patients with chronic hepatitis B (CHB) have up to a 100-fold higher risk of developing HCC, which is the fourth leading cause of cancer-related death worldwide, with ~780,000 deaths/year[1,2]. Integrated viral DNA has been detected in 85–90% of HBV-related HCCs, and its presence in tumours that develop in the non-cirrhotic livers of children or young adults further supports the role of viral DNA integration in hepatocarcinogenesis[3–7]. Integration of HBV DNA into the host genome may lead to chromosomal instability, insertional mutagenesis, deregulation of host gene expression, and production of mutant viral proteins, such as truncated surface and HBx proteins with known oncogenic properties[8,9]. Although HBV DNA insertional sites appear to be randomly distributed throughout the host genome, the recent use of next-generation sequencing (NGS) approaches has led to the identification of HBV integration enrichment into specific cancer-driver genes, including *TERT, MLL4, CCNE1* and *CCNA2*, in tumour tissues[7,10–16]. Moreover, a recent study has shown that recurrent copy number alterations in cancer-driver genes may be associated with distant viral integration[16]. Although a significant number of cases have been studied, known cancer-related genes are altered by HBV integration in only a small proportion of HBV-related HCCs. Numerous studies have also demonstrated that specific genomic elements, such as repetitive sequences, DNA sequences for non-coding RNAs, and retrotransposons, are targeted by HBV integration[7,11,12,14,17–19]. One study in Hong Kong that analysed HBV-positive HCC cell lines showed that expression of a specific chimerical HBx-LINE1 transcript has a tumour-promoting function, with a large proportion of the evaluated HCCs expressing this transcript[17]. Nonetheless, HBx-LINE1 expression was not confirmed in a large series of HBV-related HCCs from European patients[20].

Despite progress involving research on HBV DNA integration, many key aspects remain unclear. Overall, the development of alternative detection methods for HBV integration may help in gaining better insight into the mechanisms involved in the carcinogenic process induced by HBV integration.

By applying a high-throughput HBV integration sequencing (HBIS) method, in this study we detected enrichment of virus integration in mitochondrial DNA (mtDNA) from both tumour and non-tumour liver tissues of HCC patients. Moreover, we applied HBIS and RNASeq to mitochondria purified from HBV-induced HepAD38 cells and detected multiple HBV integrations into mtDNA as well as HBV-mitochondrial fusion transcripts. All mitochondrial integrations in tumour tissues were clonally expanded and involved both the oxidative phosphorylation (OXPHOS) mitochondrial genes and the *D-loop* region. We also found HBV RNA sequences to be imported into the mitochondria of hepatoma cells and that polynucleotide phosphorylase (PNPASE) might be involved in viral transcript import.

## Results

### HBV integration sequencing and identification of clonally expanded integrations

A total of 297 integration libraries were constructed from the tumour tissue of seven patients with HBV-related HCC and paired adjacent non-tumour liver tissue that was available for six of the seven patients (Table 1). The libraries were generated using a modification of the integration sequencing method developed by Cohn et al. to study the integration profile of HIV-1 in CD4 + T cells[21]. We refer to the method used in this paper as high-throughput HBV integration sequencing (HBIS) (Fig. 1).

A total of 3339 HBV integration sites were detected in the 7 patients studied (Supplementary Data 1): 2913 integrations in tumour and 426 in non-tumour tissue specimens. The average number of integration sites in tumour and adjacent non-tumour tissues was $416 \pm 387$ and $71 \pm 56$ ($P = 0.054$, Student's $t$ test), respectively. All patients studied showed HBV integration, with an average of 257 integrations per patient. In addition, HBV integration sites were detected in all tumour and non-tumour tissue specimens examined, with each site supported by at least three paired-end reads.

As demonstrated in previous studies[21–23], by coupling random DNA fragmentation (generating unique linker ligation sites) with paired-end sequencing (enabling precise identification of both the integration site and fragmented end), it is possible to identify clonally expanded integrations (identical integration sites with distinct fragmentation ends, deriving from the clonal expansion of a single integration event) and single integrations (unique integration sites with a single fragmented end). Therefore, the HBIS method enabling the enumeration of identical viral integration sites allows for the identification of clonally expanded integrations. We estimated 2107 (63%) of the 3339 detected HBV integration sites to be clonally expanded integrations. Among the 2107 clonal integrations, 1817 (86.2%) were found in tumour tissues and 290 (13.8%) in non-tumour tissues. However, the proportion of clonally expanded integration breakpoints was significantly higher in non-tumour tissues [290/426 (68%) versus 1817/2913 (62.4%); $P = 0.025$, $Z$ test] than in tumour tissue samples when the total number of integrations was taken into account. Although the size of individual clones varied from 3 to 3253 in tumours and from 3 to 376 in non-tumour tissue samples,

**Table 1 Demographic, clinical and virologic characteristics of the seven studied patients at the time of HCC surgical treatment.**

| Patient | Age | Sex | Number of tumour nodules | Size in mm of the HCC nodule | Tumour grading | Months of antiviral treatment before surgery | Serum HBV DNA (IU/ml) | Total liver HBV DNA (copies/cell) | | Liver HBV cccDNA (copies/cell) | | Liver HBV RNA (relative to G6PDH) | |
|---|---|---|---|---|---|---|---|---|---|---|---|---|---|
| | | | | | | | | T | NT | T | NT | T | NT |
| 0501 | 58 | M | 1 | 35 | G2 | No treatment | $2.1 \times 10^6$ | 60 | 5 | $0.13 \times 10$ | $4 \times 10^{-2}$ | $1 \times 10^2$ | $2 \times 10^{-1}$ |
| 0502 | 72 | M | 2 | 50* | G2 | 1 | $1.2 \times 10^2$ | 1.5 | 5.1 | $8 \times 10^{-3}$ | $1 \times 10^{-3}$ | $3 \times 10^{-1}$ | $1 \times 10^{-2}$ |
| 0503 | 74 | F | 1 | 20 | G1 | 40 | N.D. | 6.4 | 30 | $9 \times 10^{-1}$ | $2 \times 10^{-3}$ | $4 \times 10^{-1}$ | $5 \times 10^{-2}$ |
| 0504 | 75 | M | 1 | 10 | G1 | 0 | $7.7 \times 10^7$ | $1.3 \times 10^3$ | $8.3 \times 10^2$ | $5 \times 10^{-1}$ | $1.3 \times 10$ | $4 \times 10^3$ | $4 \times 10^2$ |
| 0505 | 67 | M | 1 | 10 | G1 | 159 | N.D. | 7.7 | 12 | $6 \times 10^{-3}$ | $3 \times 10^{-3}$ | $2 \times 10^{-3}$ | $1 \times 10^{-1}$ |
| 0506 | 54 | M | 1 | 15 | G1 | 0 | $3.7 \times 10^5$ | $1.9 \times 10^3$ | $1.5 \times 10^4$ | $1.6 \times 10$ | $5 \times 10^{-1}$ | $7.4 \times 10$ | $4 \times 10$ |
| 0507 | 67 | M | 1 | 20 | G2 | 9 | N.D. | 1.7 | N.A. | $1 \times 10^{-3}$ | N.A. | $6 \times 10^{-2}$ | N.A. |

*T* Tumour, *NT* non-tumour, *N.D.* not detected, *N.A.* not available.
*50 mm is the size of the larger HCC nodule of patient 0502.

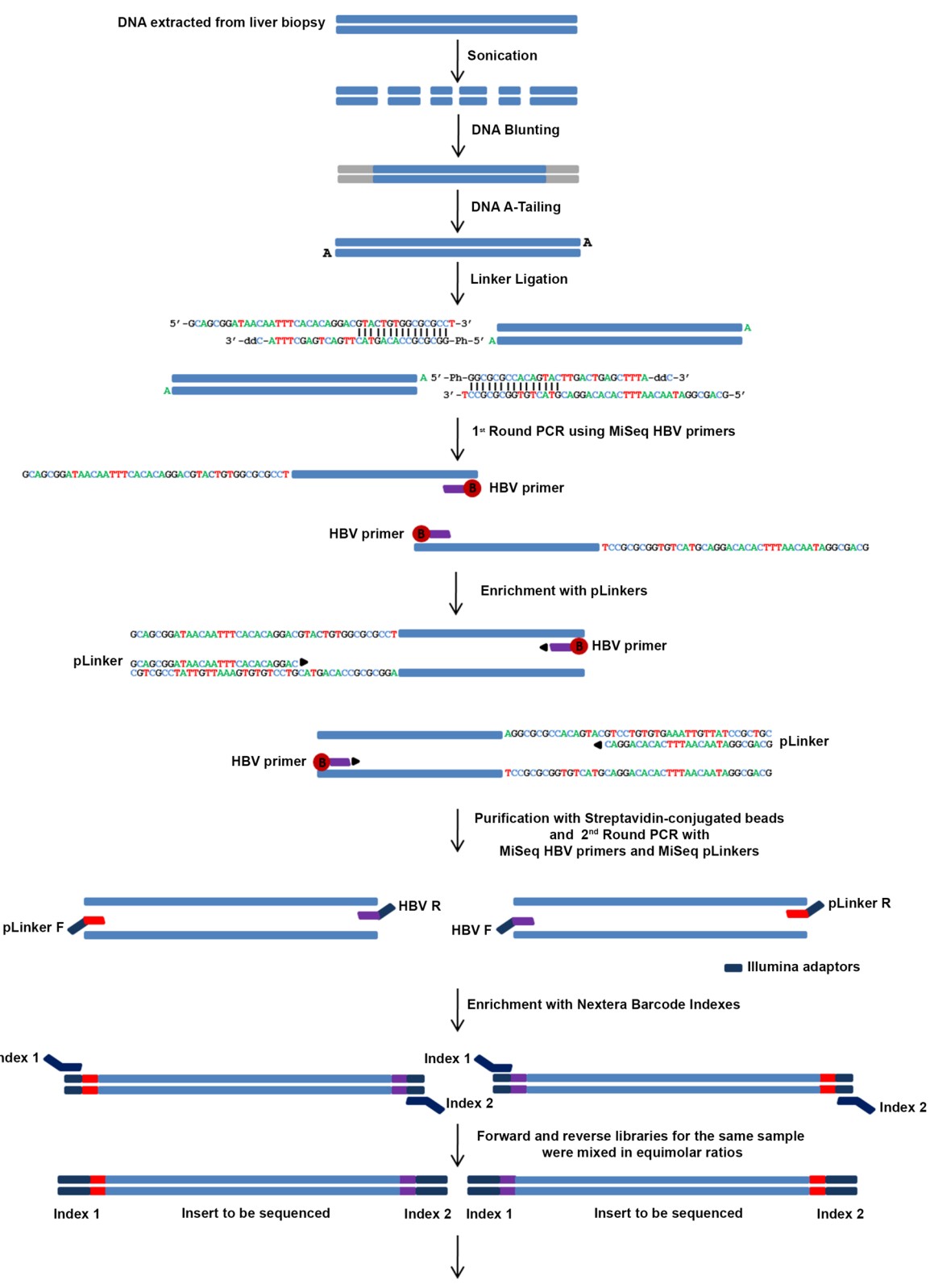

**Fig. 1 A schematic outline of library construction using the high-throughput HBV integration sequencing (HBIS) approach.** By applying HBIS, HBV integration sites were recovered by semi-nested ligation-mediated PCR from host DNA (fragmented by sonication, A-tailed, and ligated to asymmetric DNA linkers) and the use of either forward or reverse primers specific to different HBV genomic regions. Targeted sequences were further enriched by performing nested PCR with forward or reverse MiSeq HBV primers and forward or reverse MiSeq pLinkers, all containing an Illumina adaptor for flow cell surface annealing. The adaptor-ligated fragments were enriched by PCR with Illumina primers Index 1 and Index 2, and the PCR products were subjected to high-throughput paired-end sequencing. The reads were aligned to a hybrid genome including the human GRCh38.p10 (GenBank accession: GCA_000001405.25) and HBV (GenBank accession: NC_003977) reference genomes.

| Table 2 Mean amounts of total HBV DNA, cccDNA and pgRNA in matched tumour and non-tumour tissue samples. | | | |
|---|---|---|---|
| **Tissue sample** | **HBV parameters mean ± SD (copies/cell)** | | |
| | **Total HBV DNA** | **HBV cccDNA** | **pgRNA** |
| Tumour | 468.2 ± 792[*] | 2.7 ± 5.9[#] | 87.6 ± 226[$] |
| Non-tumour | 2269 ± 5622[*] | 2 ± 4.9[#] | 110.7 ± 219.7[$] |

*cccDNA* covalently closed circular DNA, *pgRNA* pregenomic RNA.
[*]$P = 0.376$, [#]$P = 0.818$, [$]$P = 0.220$ (Student's *t* test).

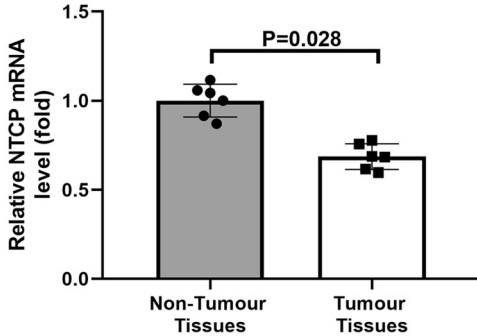

**Fig. 2 Expression profile of the HBV entry receptor NTCP in tumour and non-tumour liver tissues.** The expression of *NTCP* gene was evaluated in the six non-tumour and seven tumour liver tissues using qRT-PCR. *NTCP* was expressed in all the analysed tumours, though at lower levels than non-tumour tissues. Results are expressed as fold change relative to *NTCP* expression levels in non-tumour tissues and normalised to *GAPDH* expression. Measurements were performed in duplicate and the results are the average of three independent experiments ($n = 3$). Error bars represent the mean ± SD. Unpaired Student's *t* test, $P = 0.028$.

the mean size of clones from the former was similar to that of clones from the latter (14.8 ± 125.3 versus 18.3 ± 35; $P = 0.589$, Student's *t* test). Therefore, the tumours evaluated exhibited both a significantly higher proportion of small clones [clone size ≤100: 1795/1817 (98.8%) versus 281/290 (96.8%); $P = 0.008$, *Z* test] and a significantly higher number of single integration sites [1096/2913 (37.6%) versus 136/426 (31.9%); $P = 0.022$] than non-tumour tissues. These results are in contrast with previously reported findings[16], which may be due to the limited number of tumours studied and to the fact that all the analysed tumours were (a) histologically well-differentiated (Edmondson–Steiner grade I–II, ES GI-GII)[24], (b) showed levels of HBV replication comparable to those detected in matched non-tumour tissue samples (Table 2), and (c) expressed sodium taurocholate cotransporting polypeptide (*NTCP*), the entry HBV receptor, though at lower levels than non-tumour tissues ($P = 0.028$, Student's *t* test) (Fig. 2). Hence, by maintaining the ability to support virus replication and reinfection, well-differentiated tumour cells may be prone to the occurrence of new HBV integration events over time.

**Characteristics of HBV integration events in the human genome.** The detected HBV integration sites were randomly distributed across the entire genome, and no hotspot was revealed by Monte Carlo simulation[25]. However, analysis of the distribution of HBV breakpoints across individual chromosomes—after normalisation of the number of integrations to the length of each chromosome—showed statistically significant enrichment of integration sites in the centromeric region of chromosome 20 in tumour samples ($P = 0.018$, NPC test) (Supplementary Data 2), whereas non-tumour samples exhibited a relatively random distribution of integration sites across chromosomes. To evaluate whether the position of HBV integrations in the host genome is associated with clonal expansion, we performed a comparison between the genomic location of the clonally expanded and single HBV integrations in tumour and non-tumour tissues. We found that both clonal and single integrations showed a comparable distribution in genes and intergenic regions of the tumour and non-tumour tissue samples.

As clonal integrations have been associated with genes involved in malignant transformation[16], we examined our dataset for the enrichment of integrations in HCC-associated genes. The clonally expanded genes targeted by HBV integration from our study were compared to those recently reported in the Viral Integration Site Database (VISDB, available at https://bioinfo.uth.edu/VISDB), the most comprehensive collection of viral integration sites (VISs) for DNA viruses (including HBV) and RNA retroviruses[26]. Of the 5573 HBV-targeted genes in HCC reported in VISDB, 381 were detected in tissue specimens from the 7 studied patients (323 in tumour and 58 in non-tumour tissue samples). Among the 381 HBV-targeted genes, 35 (*AOAH, ANKS1B, CAMTA1, CDH4, CYP2E1, DSCAML1, EXOC4, FCHSD2, FSTL4, HIVEP3, HSPA12A, IGH, IQSEC1, JARID2, KCNQ3, LMF1, MED26, MLPH, MPG, NFAT5, NCOR2, PLD5, PTPRG, PTPRM,*

*TRAPPC9, SLC29A3, SDCCAG8, SMOC1, SYNGR1, TGFBRAP1, WDR66, WNK2, ZHX2, ZMIZ1, ZNF536*) were recurrently affected (in 2 or more samples) in tumour tissues and 1 (*NRG3*) in non-tumour tissues. Integration events occurring in single samples were observed in the following cancer-driver genes: *TERT, KMT2B (MLL4), RIMS1, FN1, RBFOX1, CNTNAP2* and *THRB*[7,11,12,14,27,28]. In addition, we found seven other genes recurrently affected by HBV integration in tumour tissue samples: *ECE1, EVI5L, KIF13B, MTX1P1, PLXND1, TECR* and *USP24*.

The HBV integration sites were also annotated to analyse their distribution in distinct genomic elements. Integrations were more frequent in simple repeats in non-tumour than in tumour samples [60/290 (20.7%) versus 88/1817 (4.8%); $P < 0.001$, *Z* test], whereas statistically significant enrichment of integrations was observed for repetitive sequences in tumour compared to non-tumour samples [1376/1817 (75.7%) versus 179/290 (61.7%); $P < 0.001$, *Z* test]. Tumour tissues also showed a significantly higher number of integrations in short interspersed nuclear elements (SINEs, 310/1817 (17%) versus 26/290 (8.9%); $P < 0.001$] and long terminal repeats (LTR) retrotransposons [264/1817 (14.5%) versus 28/290 (9.6%); $P = 0.024$]. A large number of HBV integrations also occurred in long interspersed nuclear elements (LINEs) [250/1817 (13.7%) versus 40/290 (13.7%); $P = 0.999$], though no significant differences were observed between tumour and non-tumour tissue samples (Supplementary Fig. 1). Interestingly, enrichment of HBV integrants, including the X genomic region, was found in the LINE sequences of tumour tissues [195/1817 (10.7%) versus 19/290 (6.5%); $P = 0.027$], whereas viral integrants, including S genomic sequences, were significantly more frequent in LINEs (52/1817 (2.9%) versus 22/290 (7.6%); $P < 0.001$) of non-tumour tissue samples.

**Integration breakpoints in the HBV genome.** Characterisation of breakpoints in the HBV genome (Supplementary Fig. 2) revealed accumulation of integration breakpoints at nucleotides 1500–2000—including the 3'-end of the *HBx* and 5'-end of the *Precore/Core* genes—in tumour tissue samples [855/1817 (47.1%) versus 75/290 (39.5%), $P < 0.0001$] compared with non-tumour tissues. In tumour tissues, a higher proportion of integration sites at nucleotides 1000–1200, including the *ENH I/X* promoter [340/1817 (18.7%) versus 20/290 (6.9%), $P < 0.0001$], was also found.

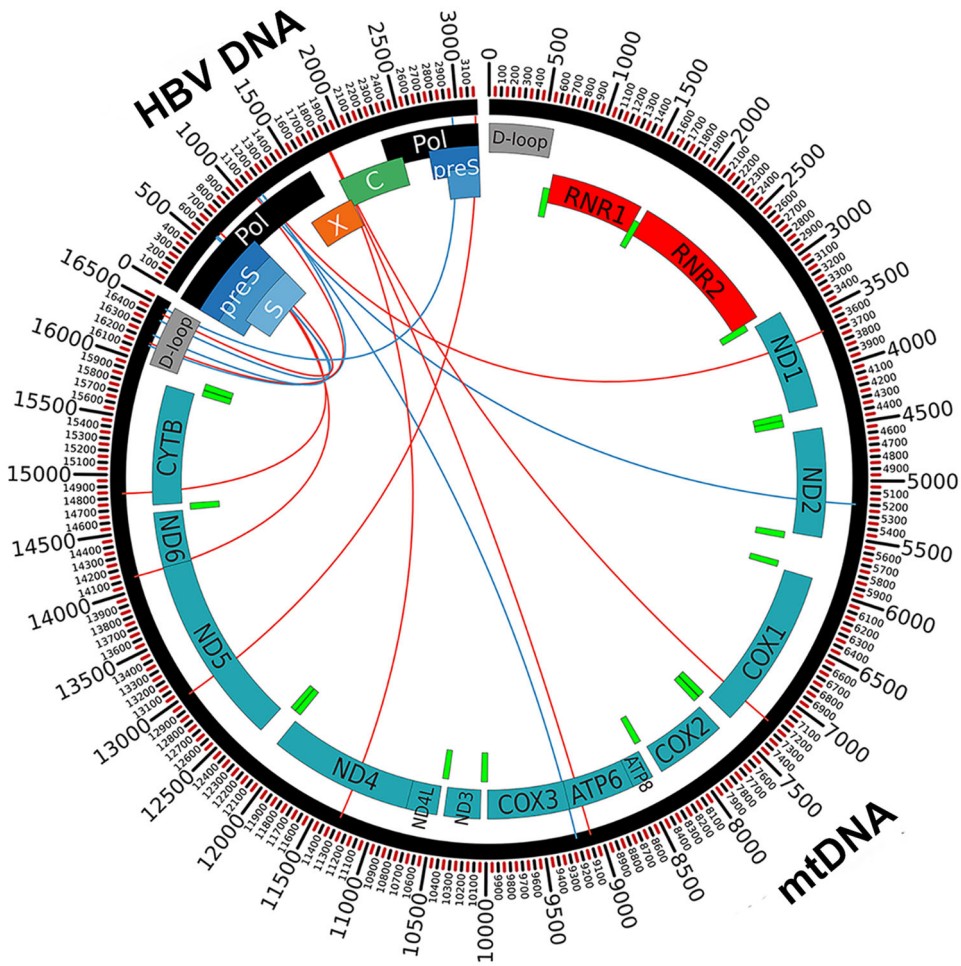

**Fig. 3 Distribution of integration sites in the HBV genome and in the mitochondrial genome represented in the circos plot.** HBV genes are represented in black (*Polymerase, Pol*), green (*Precore/Core, C*), different gradients of blue (*preS/S*), and orange (*X*). Mitochondrial genes are represented in teal, bright green, and red. The mitochondrial *D-loop* regulatory region is represented in grey. HBV-mitochondrial junctions in tumours tissues are represented by red lines. HBV-mitochondrial junctions in non-tumours tissues are represented by blue lines.

In non-tumour tissue samples, HBV breakpoints were significantly enriched at nucleotides 400–750 of the *S* gene [460/1817 (25.3%) versus 147/290 (50.7%), $P < 0.0001$]. Furthermore, the *S* gene immunodominant "a determinant" sequence was included in a significantly higher proportion of chimeric sequences from non-tumour samples [73/290 (25%) versus 75/1817 (4%); $P < 0.0001$].

To elucidate the mechanism of HBV integration, the presence of microhomology (MH) sequences between nuclear DNA and HBV DNA integrants at the level of integration sites was investigated. We found that most of the 3339 HBV integration sites in our cases showed the presence of MH sequences in both tumour and non-tumour samples (Supplementary Table 1 and Supplementary Data 1). The presence of MH sequence enrichment at the junction site between integrated HBV and cellular DNA indicates that the MH-mediated end-joining (MMEJ) DNA repair mechanism may promote viral insertion at the level of genomic breaks.

**HBV integration breakpoints in mitochondrial DNA**. Our analysis of HBV breakpoint distribution led us to identify mitochondrial DNA (mtDNA) as an HBV integration target. Indeed, compared to the number of integrations into individual chromosomes, a highly statistically significant enrichment of virus integration sites was observed in mtDNA ($P < 0.0001$, NPC test), both in tumour ($P < 0.0001$) and non-tumour samples

**Table 3 HBV integration sites located in protein-coding genes and in the *D-loop* region of mitochondria from tumour and non-tumour tissue specimens of patients 0501, 0504, 0505 and 0506.**

| Patient | Tissue sample | mtDNA coding and non-coding regions affected by HBV integration (number of integration sites) |
|---|---|---|
| 0501 | Tumour | ND2, ATP6, *D-loop* |
| 0504 | Tumour | RNR2 (3), ND1, ATP6, ND4 (2), ND5 (2), CYTB, *D-loop* (2) |
| | Non-tumour | *D-loop* |
| 0505 | Non-tumour | *D-loop* (2) |
| 0506 | Tumour | COX1, ND4 |

*mtDNA* mitochondrial DNA.

($P < 0.0001$), after normalisation of the number of integrations to the length of mtDNA (considering a median amount of 2000 copies of mtDNA per hepatocyte[29]) (Supplementary Data 2). We found 20 HBV integrations in mtDNA (ranging in length between 49 bp and 187 bp) from tissue samples obtained from 4 (0501, 0504, 0505 and 0506) of the seven studied patients (Fig. 3, Table 3 and Supplementary Data 3). Notably, we observed a correlation between HBV integration in mtDNA and higher levels of serum HBV DNA ($r_S = 0.899$, $P = 0.006$; Spearman correlation

test) as well as higher intrahepatic amounts of HBV DNA ($r_S = 0.866$, $P = 0.012$) and pgRNA ($r_S = 0.878$, $P = 0.021$) in these patients. Among the 20 HBV integration sites, 17 were detected in mtDNA from tumour tissues, and 3 were found in mtDNA from non-tumour tissue samples (Table 3 and Supplementary Data 3). All integration breakpoints detected in mitochondria from tumour and non-tumour tissues were clonally expanded integrations. HBV integration sites in mtDNA from tumour tissues, compared with those from non-tumour tissues (11/17 versus 0/3; $P = 0.073$, Fisher's exact test), were preferentially located within mitochondrial genes related to the OXPHOS system, which is the final biochemical pathway for ATP production and is composed of five multiprotein enzyme complexes (I–V) and two electron carriers (coenzyme Q and cytochrome c)[30]. In particular, HBV integrations in tumour tissue samples were detected in mitochondrial genes encoding the ND1, ND2, ND4, and ND5 core subunits of OXPHOS Complex I, CYTB of OXPHOS Complex III, COX1 of OXPHOS Complex IV, and ATP6 of OXPHOS Complex V. The *RNR2* (16 S rRNA) mitochondrial gene and the non-coding regulatory displacement loop (*D-loop*) mitochondrial region[30] were also targets of HBV in tumour tissues. In contrast, HBV integrations in mtDNA from non-tumour tissue samples were only found within the *D-loop* region (3/17 versus 3/3; $P = 0.017$). Patient 0504 showed the highest number of viral integrations in mtDNA, with 12 different integration sites in the tumour sample (3 in *RNR2*, 2 in *ND4*, 2 in *ND5*, 1 in ND1, 1 in *ATP6*, 1 in *CYTB*, and 2 in the *D-loop* region) and 1 in the non-tumour tissue sample (in the *D-loop* region) (Fig. 3, Table 3 and Supplementary Data 3). HBV integration in the *D-loop* region (Supplementary Fig. 3a, b), as well as in the *RNR2, CYTB* (Supplementary Fig. 4a, b), and *ND5* (Supplementary Fig. 5) genes, in this patient was confirmed by Sanger sequencing. Concerning the *ND5 gene*, cloning and Sanger sequencing led us to identify the entire HBV sequence integrated in this gene (Supplementary Fig. 5). The length of the integrated viral fragment was 363 bp and included two gamma interferon activation site (GAS) elements and the HBV *ENHI* region.

Analysis of HBV genome breakpoints showed non-tumour-enrichment of integration sites at nucleotides 2877–752 (including the *preS/S* region) within the *D-loop* region and a higher proportion of breakpoints at nucleotides 1000–1887 (including the *ENH I/X* promoter, the *X* gene, the *ENH II/BCP*, and the Precore region) within protein-encoding mitochondrial genes (Fig. 3) in tumour tissues. The presence of microhomology (MH) ranging from 1 bp to 18 bp between integrated HBV and mtDNA at the site of the virus-mitochondrial junction was observed for 19 of the 20 integration junctions (Supplementary Data 3), suggesting that MMEJ may also play a major role in the process of HBV integration into mtDNA.

To verify the effect of such viral integration on mitochondrial gene expression, we performed RNASeq analysis, though high-quality RNA (RNA Integrity Number ≥8) to perform this analysis was only obtained from patient 0504. Among the chimeric transcripts detected in the tumour tissue, we identified fusion sequences containing part of the *RNR2* and *ND4* mitochondrial genes, which had corresponding integration sites in mtDNA (Supplementary Data 3).

It is known that mtDNA can enter the nucleus and integrate into the nuclear genome at DNA double-stranded breaks (DSBs), giving rise to nuclear copies of mtDNA sequences (*numts*)[31]. Therefore, we cannot exclude that the HBV integrants detected by HBIS in the mitochondria from the patients might instead be located within *numts*. In an attempt to clarify this, we conducted a deeper investigation of HBV integration in HepAD38 cell line, which supports tetracycline (Tet)-off inducible HBV replication[32]. Upon Tet removal from the culture medium, these

cells produce viral RNAs, accumulate subviral particles in the cytoplasm, which contain DNA intermediates characteristic of viral replication, and secrete virus-like particles[32] containing either HBV DNA or RNA into the supernatant[33]. We investigated HBV integration in HepAD38 cells—by both HBIS and RNASeq—after Tet removal for 7 days. At this time point, mean amounts of HBV DNA and HBV RNA in HepAD38 cells were $1.1 \times 10^3 \pm 6.0 \times 10^2$ and $1.0 \times 10 \pm 5.9 \times 10^{-1}$ copies/cell, respectively; while mean amounts of HBV DNA and HBsAg in the cell supernatants were $2 \times 10^6 \pm 2.7 \times 10^4$ copies/mL and $4.1 \times 10^3 \pm 7.2 \times 10^2$ IU/mL, respectively. Therefore, at this time point, we applied HBIS to investigate the presence of HBV integration sites both in DNA extracted from purified nuclei and in DNA obtained from purified mitochondria of HBV-producing HepaD38 cells, whereas we applied RNASeq to analyse the presence of hybrid transcripts in purified nuclei plus cytoplasm and in isolated mitochondria from the same cells. Based on HBIS, no viral integration was detected in potential *numts* present in nuclear DNA from HBV-producing HepAD38 cells, whereas we did detect HBV integration sites in the *COX1*, *RNR2* and *ND2* genes in DNA obtained from purified mitochondria (Supplementary Data 4). Analogously, RNASeq revealed no viral integration in nuclear/cytoplasmic RNA extracts from HBV-producing HepAD38 cells, but several HBV-mitochondria chimeric sequences containing part of the *COX1*, *COX3*, *ND4* and *ND6* sequences were identified in the RNA isolated from mitochondria of HBV-producing HepAD38 cells (Supplementary Data 4 and Supplementary Fig. 6).

**HBV RNA is imported into mitochondria**. Liver tissue specimens for additional analyses were only available from three (0501, 0504 and 0505) of the four patients harbouring HBV integration in mtDNA. Thus, to further investigate HBV integration into mtDNA, the presence of the free HBV genome and HBV transcripts was analysed in mitochondria and nuclei from tumour tissue specimens of patients 0501, 0504 and 0505 and from liver tissue specimens of five patients with HBeAg-negative chronic hepatitis B (CHB) with active viral replication (mean level of serum HBV DNA: $5.6 \times 10^7 \pm 3.06 \times 10^7$ IU/mL) (Fig. 4a, c). Mitochondria and nuclei from the liver tissue specimens of eight HBV-negative patients and from HepG2 cells were used as negative controls (Fig. 4b, d). Neither the full-length HBV genome nor HBV cccDNA was detected in mitochondria from any of the liver tissue specimens analysed or HepG2 cells (Fig. 4a, b). However, analysis of HBV transcripts revealed the presence of HBV RNA sequences corresponding to the *PreS1, S, Epsilon* and *X* genomic regions in mitochondria from all HBV-positive liver tissue specimens but not in mitochondria from the HBV-negative liver tissues or HepG2 cells (Fig. 4e). It is known that mitochondria import a wide variety of nucleus-encoded RNAs[34–38]. Import of most of these RNAs into mammalian cells is regulated by PNPASE, which also possesses $3' \rightarrow 5'$ exoribonuclease and poly-A polymerase activities and localises to the mitochondrial intermembrane space (IMS)[35,36,39]. Considering that PNPASE translocates into mitochondria various RNAs containing a distinct import signal, namely, a small stem-loop structure[35], we sought to determine whether a similar import sequence (and structure) is present in HBV transcripts. Notably, an almost identical import signal sequence was identified in the *preS1* region (nt 3049–3065) (Fig. 5a). Furthermore, we wondered whether the stem-loop structures of HBV transcripts at the 5' end and near the 3' end, such as "epsilon" and PRE [post-transcriptional regulatory element, which includes the stem loop α (*SLα*, nucleotides 1292–1321) and *SLβ* (nucleotides 1411–1433)][40], function as mitochondrial import sequences. After ascertaining that PNPASE was produced by HepG2 cells (Fig. 5b), we verified whether HBV

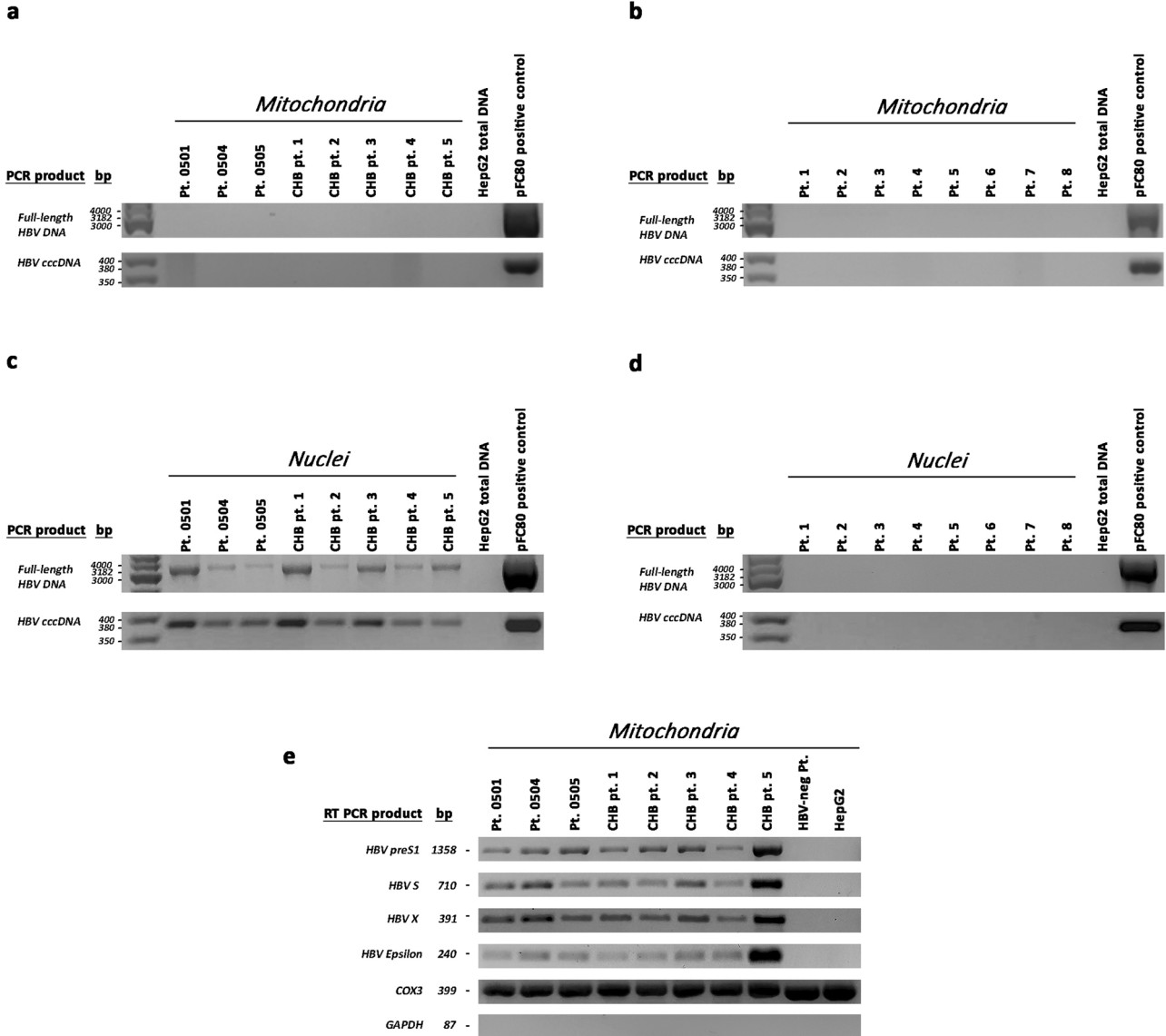

**Fig. 4 HBV RNA sequences, but not full-length HBV DNA or cccDNA, were detected in mitochondria from HBV-positive liver tissue specimens.** Full-length HBV genome, HBV cccDNA and HBV RNA in mitochondria were analysed in nuclease-treated mitoplasts and in nuclei isolated from tumour tissue specimens obtained from patients 0501, 0504 and 0505 and from liver tissue specimens obtained from 5 chronic hepatitis B patients (CHB) with active viral replication. Nuclease-treated mitoplasts and nuclei isolated from the liver tissue specimens obtained from eight HBV-negative patients (Pt.1–Pt.8) were used as negative controls. Mitoplasts with the mitochondrial outer membrane removed by digitonin treatment were used to avoid cytosolic contamination. **a** Absence of full-length HBV genome and HBV cccDNA in mitochondria isolated from liver tissue specimens of HBV-positive patients. **b** Absence of full-length HBV genome and HBV cccDNA in mitochondria isolated from liver tissue specimens of HBV-negative patients. **c** Detection of full-length HBV genome and HBV cccDNA in nuclei isolated from liver tissue specimens of HBV-positive patients. **d** Absence of full-length HBV genome and HBV cccDNA in nuclei isolated from liver tissue specimens of HBV-negative patients. Total DNA extracts from HepG2 cells and the recombinant plasmid pFC80 (containing 4 head-to-tail 1.0× length HBV genomes) were used as a negative and positive control, respectively, of PCR experiments. **e** Detection of HBV RNA sequences corresponding to the *PreS1*, *S*, *X* and *E epsilon* genomic regions in mitochondria isolated from the HBV-positive liver tissue specimens but not in mitochondria from the HBV-negative liver tissue or HepG2 cells. *GAPDH* was used as a cytosolic marker and *COX3* as a mitochondrial marker. *COX3* mRNA, but not *GAPDH* mRNA was detected in mitochondria from the same HBV-positive and HBV-negative liver tissue samples or HepG2 cells.

RNAs also localise to mitochondria by performing an in vitro import assay using a synthesised biotin-labelled sequence of the *PreS1* transcript (nt 2850–837, including the putative *PreS1* import signal), of the Polymerase transcript (nt 670–1350, including *SLα*), of the pgRNA (nt 1781–2068, including epsilon), and of the *X* transcript (nt 1376–1840, including *SLβ*). RNA sequences and mitoplasts were isolated from HepG2 cells. Prior to RNA isolation and RT–PCR, nuclease treatment was performed to remove non-imported RNA. All biotin-labelled HBV RNA sequences were imported into mitochondria, but not *COX3* mRNA that was used

as a control (Fig. 5c, d). These results led us to verify whether PNPASE may help import HBV RNAs. Thus, we investigated whether HBV RNAs directly bind to PNPASE. Mitochondria isolated from HepG2 cells—after treatment with digitonin and nuclease—were incubated with in vitro-synthesised, biotin-labelled HBV RNAs. The biotinylated viral RNA sequences were pulled down using streptavidin-coated beads, and the RNA-binding PNPASE was detected by western blot analysis (Fig. 6). Importantly, PNPASE, but not HSP90 (used as control), bound to the in vitro-transcribed RNA sequences containing the putative *PreS1*

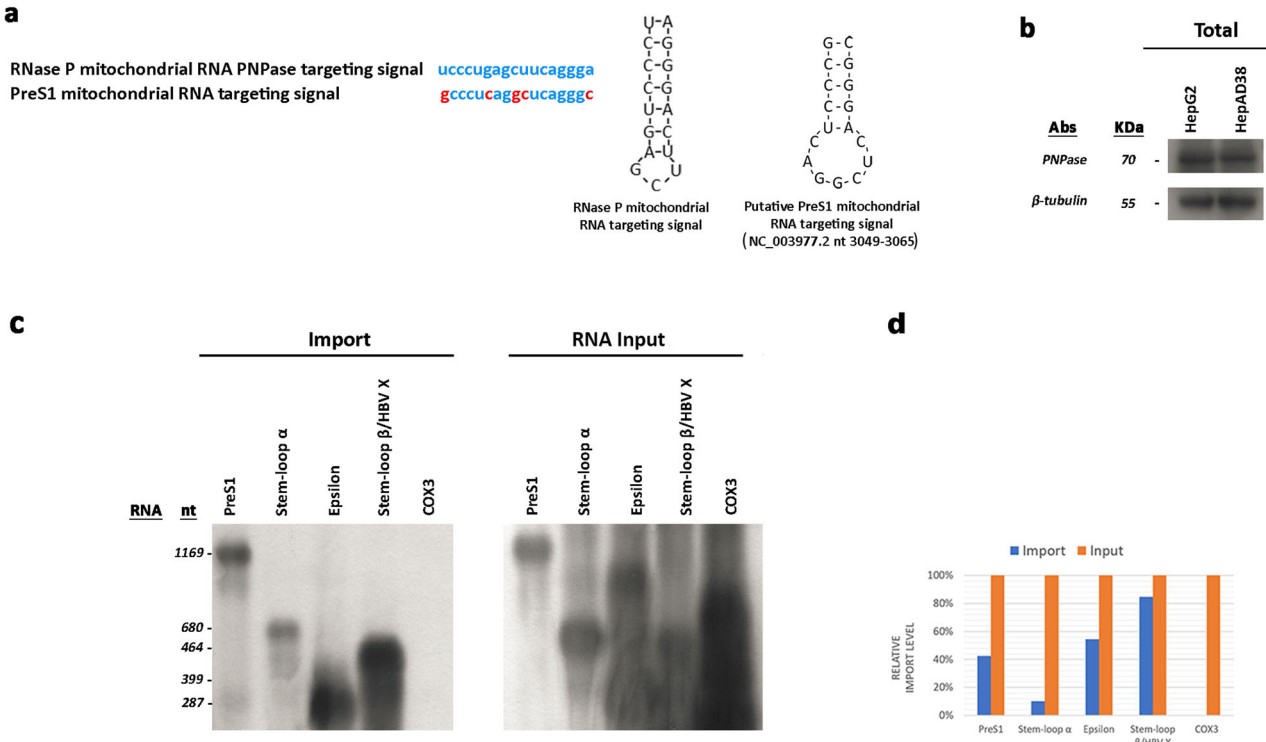

**Fig. 5 HBV RNA sequences are imported into mitochondria. a** Alignment of nucleotide sequences and secondary structures of mitochondrial RNA targeting signals in *RNase P* and *HBV PreS1* RNAs (GenBank reference sequence: NC_003977; HBV genome nucleotides: 3049–3065). **b** PNPASE immunoblot from HepG2 and HepAD38 cells. β-Tubulin was used as a loading control. **c** In vitro import of HBV RNA fragments corresponding to *PreS1* (including the putative PreS1 targeting signal), *Polymerase* (*Pol*) (including *Stem-loop α, SLα*), *pgRNA* (including *epsilon*) and *X* (including *Stem-loop β, SLβ*) transcripts as well as of a *COX3* mRNA fragment into mitochondria from HepG2 cells. In vitro-transcribed *PreS1, Pol-SLα, epsilon* and *X–SLβ*, or *COX3* RNA (Input) were incubated with mitochondria from HepG2 cells. Not-imported RNA was digested with nuclease. **d** Quantification of the relative import level by densitometry measurements.

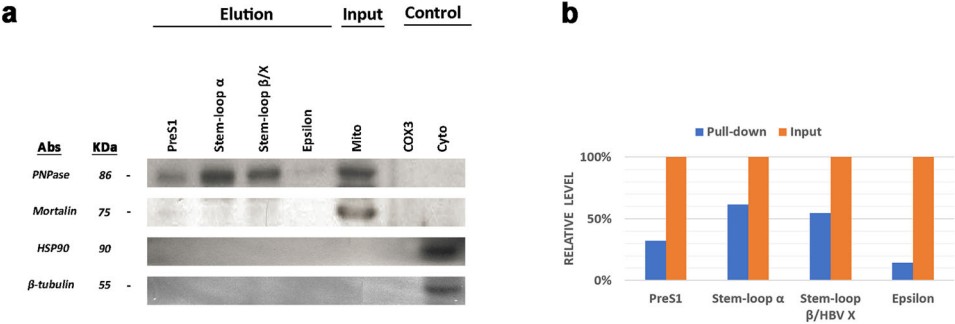

**Fig. 6 HBV RNA sequences bound specifically to PNPASE.** Mitochondria isolated from HepG2 cells—after treatment with digitonin and nuclease—were incubated with in vitro-synthesised, biotin-labelled HBV RNAs. The biotinylated viral RNA sequences were pulled down using streptavidin-coated beads, and the RNA-binding PNPASE was detected by western blot analysis. **a** PNPASE, but not HSP90 (used as control), bound to the in vitro-transcribed RNA sequences containing the putative *PreS1* targeting signal (nucleotides (nt) 2850–837), *Stem-Loop α* (nt 670–1350), *epsilon* (nt 1781–2068) and *Stem-loop β/ X* (nt 1376–1840). Moreover, HBV RNA sequences bound specifically to PNPASE, though the *COX3* mRNA sequence did not. β-Tubulin was used as a cytosolic marker and Mortalin as a mitochondrial marker. **b** Quantification of the relative HBV RNA-binding PNPASE levels by densitometry measurements.

import signal (nt 2850–837), *SLα* (nt 670–1350), epsilon (nt 1781–2068) and *SLβ* (nt 1376–1840) (Fig. 6a, b). Moreover, HBV RNA sequences bound specifically to PNPASE, though the *COX3* RNA sequence did not (Fig. 6a, b). These results implicate the structural specificity of mitochondrial viral RNA import and direct involvement of PNPASE in this process.

## Discussion

In this study, we developed a high-throughput HBV integration sequencing (HBIS) method that allows for sensitive identification of HBV integration sites and enumeration of integration clones. The HBIS approach was adapted from an integration sequencing method originally developed to study the integration profile of HIV-1 during latent and active infection[21]. This approach led us to identify and characterise a very large number of HBV integration sites in tumour and adjacent non-tumour liver tissues from patients with HBV-related HCC. All tumour and non-tumour tissues showed HBV integration, with an average of 257 integrations per patient. In accordance with previous studies[7,11,27,41], we found the total number of HBV integration sites and the number of clonally expanded integrations to be

higher in tumour compared to non-tumour liver tissues. However, tumours showed a significantly higher proportion of small clones and single integration sites than non-tumour tissues. These findings may be because the analysed HCCs were histologically graded as well-moderately differentiated, supported HBV replication and transcription, and expressed the NTCP viral entry receptor, characteristics that may have rendered these HCCs susceptible to accumulation of integrations at a higher frequency than adjacent non-tumour tissues. The persistence of HBV replication and the occurrence of new integration events in tumour tissues may also explain the large number of HCC-associated genes targeted by HBV integration that were found in each single tumour. Many of these integrations may have occurred in already-dysregulated cancer genes; thus, they may merely represent an epiphenomenon and act as passenger events. Nevertheless, some of these integrations might still have a role in HCC progression. These results suggest that tumour differentiation and its permissive state to HBV replication should be taken into account when evaluating HBV integration in HCC. Indeed, such aspects may greatly influence the number and type of HBV integration events among distinct HCCs, leading to biased results when large series of tumours with different degrees of differentiation are evaluated and compared with paired non-tumour liver tissues. Moreover, in accordance with previous reports, we observed important enrichment of HBV integrations into repetitive sequences[9,14,16]. We also confirmed that integration events occur more frequently in simple repeats in non-tumour tissues[16] and that in tumour tissues, viral integration sites are often within transposable elements[17,18,42], which are known to be overexpressed and become mobilised in numerous human cancers[43,44].

Interestingly, by applying the HBIS approach and a specific bioinformatic pipeline, we identified mtDNA as a site of HBV integration. Indeed, significant enrichment of clonally expanded viral integration was found in mtDNA from both tumour and non-tumour liver tissue specimens from four of the seven studied patients. Considering (a) viral integration into mtDNA is a de novo somatically acquired mutation, (b) mtDNA is present in thousands of copies per hepatocyte, and (c) both wild-type and mutant mtDNA molecules can co-exist in a single cell (mtDNA heteroplasmy)[29], we hypothesised that HBV integration in these four patients should have occurred at a fairly constant and high rate per mitochondrial genome replication throughout many cell divisions. Remarkably, HBV integration sites in mtDNA from tumour tissue specimens were located both in the D-loop region (the major control site for mtDNA expression, containing the leading-strand origin of replication and the major promoters for transcription[45]) and in mitochondrial genes coding for proteins of the OXPHOS system, which sustains mitochondrial function and plays a central role in cellular energy metabolism[30], whereas integrations in non-tumour tissues were only detected in the regulatory D-loop region. These findings led us to suppose that HBV integration into mitochondrial OXPHOS genes might be detrimental to the survival of non-transformed hepatocytes. In fact, although a critical number of mutant mtDNAs must be present before cells are critically damaged, and tissue dysfunction as well as clinical signs become apparent (the so-called threshold effect), tissues with a high requirement for oxidative metabolism, such as muscle, the brain, and the liver, have relatively low thresholds and are particularly vulnerable to mtDNA mutation[46]. Conversely, liver cancer cells appear to be able to tolerate HBV integration in genes of the OXPHOS system. These results are in accordance with literature data demonstrating that aberrations involving OXPHOS genes are markedly enriched in several cancer types[47]. Indeed, a vast accumulation of mutated mitochondria has been shown in kidney, colorectal and thyroid cancers[47]

indicating a functional association between mitochondrial inactivation and tumorigenic effects. Mutations in OXPHOS genes are associated with an upregulation of glycolysis and lactate production, as well as with ROS and oncometabolites overproduction[48], which stimulates oncogenic and metabolic signalling pathways and impacts the tumour microenvironment[48]. This modified microenvironment may in turn induce mitochondrial metabolic reprogramming of cancer cells[49]. The metabolic plasticity of neoplastic cells conferred by mtDNA mutations is needed to satisfy cellular biomass and energetic demands required for their proliferation and for their adaptation to novel microenvironments of primary tumour and metastatic niches[49,50]. Therefore, it might be hypothesised that changes in OXPHOS genes induced by HBV integration might persist and/or be selected to shape the metabolic efficiency of preneoplastic hepatocytes or of cancer liver cells during tumour evolution.

Concerning the mitochondrial D-loop region, our results demonstrate that this region can be affected by HBV integration both in tumour and non-tumour liver tissues, indicating that viral integration in the D-loop region is an event that may precede hepatocellular transformation and might go through a positive selection during hepatocarcinogenesis. HBV integration in the D-loop region may impair replication and transcription of mtDNA and may lead to mitochondrial instability and malfunction[51]. Overall, our findings are in accordance with data from previous studies that report a high frequency of D-loop mutations both in HCC and adjacent non-HCC tissues[52–54]. Therefore, our data along with findings from previous studies[52–54] support the critical contribution of mtDNA D-loop region mutations in hepatocarcinogenesis. Results from our study also highlight a distinct distribution pattern of HBV integration sites in mtDNA in tumour and non-tumour liver tissues. However, to note is the fact that this study only evaluated a small number of cases, and the results need to be confirmed in larger cohorts of patients.

Based on RNASeq analysis, we found that the site of HBV insertion into mtDNA may be transcriptionally active. Indeed, HBV-mitochondrial fusion transcripts containing part of mitochondrial sequences of OXPHOS genes were detected both in patients and in HBV-induced HepAD38 cells. We utilised these cells for a deep investigation of HBV integration into mtDNA and to evaluate whether HBV integration can occur in numts instead of in mtDNA. It is well documented that fragments of mitochondrial DNA can translocate into the nuclear genome to generate numts[31]. However, since their description in 1983, only slightly more than 1000 numts have been identified in humans, showing very low frequencies across tested samples. Numts range in size from ~50 bp to >15 kb, with more than 20% having a length shorter than 100 bp[31,46,55]. Considering that mtDNA averages several thousand copies per hepatocyte compared to the two copies of numts in nuclear DNA (nDNA), by isolating mitochondria from cells, it is possible to completely dilute out numts, leading to numt-free mtDNA sequences. Therefore, to study mtDNA HBV integration, we isolated nuclei, cytoplasm, and mitochondria from HBV-producing HepAD38 cells and applied both HBIS and RNASeq. According to HBIS, several HBV integration sites were identified in DNA isolated from mitochondria, whereas no integration was detected in numts from nDNA. Furthermore, RNASeq revealed the presence of chimeric HBV-mitochondrial transcripts within mitochondria but not in cytoplasm or nuclei of HBV-producing HepAD38 cells, and mtDNA insertion sites may be transcriptionally active in these cells. Both HBIS and RNAseq analysis also revealed that MMEJ have a major role in HBV integrations occurring in mitochondrial genomes. Therefore, taken together, our data clearly demonstrate

that HBV can integrate into mtDNA of tumour and non-tumour hepatocytes. Some previous studies have reported data concerning HBV integration in mtDNA[16,56–60]. All these studies have utilised high-throughput HBV genome-enrichment sequencing approaches to study HBV integration, and most of them have analysed hepatoma cell lines stably expressing HBV DNA[56–59]. To the best of our knowledge, only one[16] of the papers has reported data on HBV integration in mtDNA from human liver tissues. In particular, in the supplementary dataset of this paper, 58 different HBV integration sites in mtDNA from tumour and/or non-tumour liver tissue specimens of 11 patients with HBV-related HCC have been listed[16]. The mitochondrial genomic regions most frequently targeted by HBV integration in the 11 patients were the *D-loop* region, *ND4*, *ND5*, *RNR2*, *CYTB*, *ND6*, *ND1*, *ND2* and *COX3* genes[16]. HBV integration events have also been described in mtDNA from humanised-liver tissue samples of chimeric mice[60]. In this study, Furuta et al.[60] have identified 50 distinct HBV integration sites in mtDNA from chimeric mice. These integrations (a) have been associated with higher levels of HBV replication, (b) occurred at higher frequency in the *D-loop* region, and (c) appeared to rely on MMEJ[60]. No detailed information on virus-mtDNA junctions has been provided in studies performed on PLC/PRF/5 cell lines[56,59]. However, the fact that HBV integration in mtDNA may occur in these cells—which do not replicate HBV and only express multiple distinct viral RNAs from HBV integrants[56,59]—suggests that viral RNA might be involved in the process of HBV integration in mtDNA. Despite a number of studies documenting interaction between HBV proteins and mitochondria and consequent alteration of mitochondrial functions[61–63], whether HBV nucleic acids may translocate into mitochondria has only minimally been addressed. Based on our results, HBV transcripts, but not viral full-length genome or cccDNA, can localise to mitochondria. In addition, PNPASE, a mitochondrial protein considered the first RNA import factor for mammalian mitochondria[35,36,39], possibly mediates viral RNA delivery into the mitochondrial matrix. A PNPASE-dependent RNA import sequence that we identified for the *preS1* transcript as well as known stem-loop structures specific to HBV transcripts appear to mediate mitochondrial targeting of viral RNAs. Localisation of HBV transcripts to mitochondria leads us to hypothesise that viral RNA may represent a possible substrate for HBV integration in mtDNA. The mitochondrial genome is more prone to damage and double-strand break (DSB) formation than the nuclear genome due to frequent exposure to the ROS generated by mitochondrial oxidative phosphorylation and the lack of protective histones. Considering that several reports have shown that RNA molecules can directly act as a template for the repair of mitochondrial DSBs in human cells[64], it is tempting to speculate that viral exploitation of this pathway may lead to HBV sequences being inserted into the mitochondrial genome. In summary, we found that HBV may integrate into mtDNA, with tumours and non-tumour liver tissues showing distinct profiles of viral integration into the mitochondrial genome. Moreover, our results indicate that HBV RNA may be actively imported into mitochondria and that viral RNA sequences might be involved in the process of HBV integration into mtDNA. In spite of the relatively limited sample of patients, this study offers new insight into the HBV-hepatocyte interaction and provides a new basis for investigative analyses that may lead to further comprehension of the mechanisms by which HBV insertion can drive HCC development and progression.

## Methods

**Patients**. Tumour tissues from seven HBsAg-positive patients (6 men, 1 woman; mean age 66.7 ± 8 years) with HBV-related HCC and paired adjacent non-tumour tissues from six of them were studied. The clinical, histological, and virologic characteristics of the patients are summarised in Table 1. Two normal liver samples from HBV-negative patients who underwent surgery for liver metastasis were used as a negative control. All liver tissue samples obtained by surgical resection at the Unit of Oncology Surgery of the University Hospital of Messina, Italy, were immediately frozen in liquid nitrogen and stored at −80 °C. Furthermore, in order to study the presence of free HBV genome and HBV transcripts in mitochondria, frozen liver biopsy specimens from five patients with HBeAg-negative chronic hepatitis B (CHB) (2 women, 3 men; mean age: 59.4 ± 15.8 years) showing active viral replication (mean level of serum HBV DNA: $5.6 \times 10^7 \pm 3.06 \times 10^7$) were investigated. Moreover, eight frozen liver biopsy specimens from HBsAg-negative patients (4 women and 4 men; mean age, 53.75 ± 11.5 years) with chronic liver disease (CLD) were included in this study as a control group. Two of the eight patients had an HCV-related CLD; among the six anti-HCV-negative individuals, five had nonalcoholic steatohepatitis and one had cryptogenic liver disease. The eight patients were negative for occult HBV infection. All the aforementioned liver biopsy specimens were infected with HBV genotype D. None of the patients consumed alcohol or were infected with hepatitis delta virus or HIV. The collection and processing of all samples were approved by the Ethics Committee of the University Hospital of Messina (protocol number 64/15) and conducted in accordance with the ethical principles of the Declaration of Helsinki. All individuals provided written informed consent.

**DNA and RNA extraction from liver specimens**. Extraction of total DNA and RNA from liver biopsy specimens was performed as previously described[65]. Briefly, cryopreserved liver tissue specimens from individual patients were homogenised using a TissueRuptor instrument (Qiagen, Milano, Italy) in 500 μL homogenisation buffer (50 mM Tris-HCl, pH 8.0, 1 mM EDTA, 150 mM NaCl) at 4 °C and then divided into two equal parts: one was used for DNA extraction and the other for RNA extraction. Total liver DNA was extracted from one part of the homogenate in 150 mM NaCl, 50 mM Tris-HCl (pH 7.4), 10 mM EDTA, 1% SDS, and proteinase K (800 μg/mL) overnight at 37 °C. After extraction with phenol–chloroform, nucleic acids were precipitated using pure cold ethanol, resuspended and digested with pancreatic RNase (100 μg/mL), followed by extraction with phenol–chloroform and reprecipitation in pure cold ethanol. The DNA was resuspended in 10 mM Tris-HCl (pH 7.4) and 1 mM EDTA. Total liver RNA was extracted from the other half of the liver tissue homogenate using TRIzol reagent (Invitrogen, Waltham, MA, USA) as recommended by the manufacturer. The RNA samples were treated with RQ1 RNase-Free DNase (Promega) for 30 min at 37 °C and stored at −80 °C until use. The RNA and DNA concentrations were measured at 260 nm using an ND-1000 spectrophotometer (NanoDrop Technologies).

**Quantitative real-time PCR (qRT-PCR)**. Quantification of total intrahepatic HBV DNA and cccDNA was performed using the method described in ref. [66], with minor modifications. Briefly, qRT-PCR experiments to evaluate the amounts of total intrahepatic HBV DNA or cccDNA were performed using a Light Cycler v2.0 (Roche Diagnostics) in a 20-μL reaction volume containing 100 ng DNA, 3 mmol/L MgCl₂, 0.5 μmol/L specific forward and reverse primers (Supplementary Table 2), and 0.75 μmol/L 5'-FAM-labelled and 3'-TMR-labelled TaqMan probe (Supplementary Table 2). Serial dilutions of a plasmid containing a monomeric HBV insert (Alfa Wasserman) were used as quantification standards. Amplification of total intrahepatic HBV DNA was performed under the following conditions: 95 °C for 10 min, followed by 50 cycles of 95 °C for 10 s, 57 °C for 30 s and 72 °C for 10 s, and 40 °C for 2 min. Before qRT-PCR of cccDNA, aliquots of the liver DNA extracts were treated for 2 h at 37 °C with 10 U of plasmid-safe DNase (Epicentre, Madison, WI). Amplification of cccDNA was then performed as follows: 95 °C for 10 min, 50 cycles of 95 °C for 10 s, 62 °C for 30 s and 72 °C for 20 s, followed by 40 °C for 2 min. To normalise the number of viral genomes present in each liver sample, the number of haploid genomes was evaluated using a β-globin gene kit (Light Cycler-Control Kit DNA; Roche Diagnostics). For HBV RNA analysis, 2 μg DNase-treated RNA was reverse transcribed and amplified using the SuperScript First-Strand, Synthesis System for RT–PCR (Invitrogen). Two microlitres of cDNA was quantitated by qRT-PCR (Light Cycler; Roche Diagnostics) using primers and probes designed to specifically quantify HBV pregenome RNA (pgRNA) (Supplementary Table 2). For quantification of *NTCP* mRNA, cDNA synthesis was performed using the SuperScript First-Strand, Synthesis System for RT–PCR (Invitrogen). Quantitative PCR reactions were prepared with SYBR Green (SsoFast EvaGreen supermix, Bio-Rad) according to the manufacturer's protocol. Reactions were run on a Light Cycler thermal cycler (Roche Diagnostics) with the oligonucleotide primers reported in Supplementary Table 2. The h-*G6PDH* housekeeping gene Light Cycler set (Roche Diagnostics) was used to normalise the RNA samples. All experiments were performed in duplicate and repeated three times.

**Cell lines**. The PLC/PRF/5 human hepatoma cell line (SIGMA, catalogue number 85061113, Lot Number: 10D004) containing multiple integrated HBV DNA fragments, the HepG2 cell line (ATCC, catalogue number HB-8065™), and the Vero cell line (provided by Prof Maria Teresa Sciortino) were maintained in Dulbecco's modified Eagle's medium (DMEM) with GlutaMAX, 10% FBS, 200 units/mL penicillin, and 200 μg/mL streptomycin. The HepAD38 cell line (kindly provided

by Dr. Cristopher Seeger), which is derived from HepG2 cells and contains the HBV genome (subtype ayw) under a tetracycline inducible promoter[32], was cultured in DMEM/F12 with GlutaMAX, 10% FBS, 200 units/mL penicillin, 200 μg/mL streptomycin, and 0.3 μg/mL tetracycline. In HepAD38 cells, tetracycline removal induces the production of HBV RNA and secretion of virus-like particles into the medium[32]. PLC/PRF/5, HepG2, HepAD38 and Vero cell lines tested negative by Hoechst 33258 staining for mycoplasma contamination.

**Genomic DNA extraction, sonication, linker ligation and paired-end library preparation**. For the identification of HBV DNA integration, we modified an integration sequencing method previously described to study HIV integration[21]. Genomic DNA was obtained from all tissue specimens through standard proteinase K lysis and phenol–chloroform extraction, as described above. To determine the number of liver cells in each tissue sample, we quantified the beta-globin gene using a Light Cycler v2.0 and Light Cycler-Control Kit DNA (both Roche Diagnostics). For each sample, genomic DNA isolated from 30 million cells was fragmented by sonication at 30% power for three cycles (15″on/15″off) with a SONOPULS ultrasonic homogeniser (Bandelin) to yield a 100–1000-bp distribution of DNA fragments. The DNA was then divided into 6 aliquots of 5 μg each in microcentrifuge tubes, and subsequent reactions were performed individually on these 5-μg aliquots. DNA was blunted using End-It DNA Repair Kit (Epicentre) and purified with MinElute Reaction Clean-up (Qiagen). An adenosine tail was added to the blunted DNA using 5 μL NEB buffer 2 (10×), 1 μL dATP (10 mM) and 2 μL Klenow fragment 3- > 5 exo⁻ (5000 U/mL) (New England Biolabs, Ipswich, MA) for 1 h at 37 °C. All reactions were purified by a MinElute Reaction Clean-up kit (Qiagen). Each aliquot of blunted, A-tailed DNA fragments was then ligated to 200 pmol annealed linkers (LinkerTop + LinkerBottom) (Supplementary Table 2) with 4 μL pLinker, 5 μL NEB T4 DNA ligase buffer and 1 μL T4 DNA ligase (2 × 10⁶ U/mL, high concentration) (New England Biolabs) for 1 h at 25 °C and then overnight at 16 °C. The ligase was inactivated by incubation at 70 °C for 20 min, and the reactions were purified using a MinElute Reaction Clean-up kit (Qiagen). Finally, all six reactions were pooled, and the pooled linker-ligated DNA was aliquoted into two equal parts to perform semi-nested ligation-mediated PCR with forward or reverse HBV primers (Fig. 1 and Supplementary Table 2). The forward and reverse enrichment sequences were kept separate throughout the remainder of the protocol. The DNA was divided into 1-μg aliquots; each aliquot was mixed with 20 μL Phusion HF buffer (5×), 3 μL dNTPs (10 mM), 1 μL biotinylated forward (20 μM) or reverse HBV primer (Supplementary Table 2) (2.5 μM), 1 μl Phusion Taq (2000 U/ml) (New England Biolabs) and H₂O to 50 μL. Single-primer PCRs were performed as follows: 98 °C for 1 min; 12 cycles of 98 °C for 15 s, 65 °C for 30 s and 72 °C for 45 s; 72 °C for 1 min); and a hold at 4 °C. Each tube was then spiked with 1 μL pLinker (Supplementary Table 2) (2.5 μM) and subjected to additional cycles of PCR, as follows: 98 °C for 1 min; 35 cycles of 98 °C for 15 s, 65 °C for 30 s and 72 °C for 45 s; 72 °C for 5 min; and a hold at 4 °C. Forward and reverse PCRs were purified using the QIAquick PCR purification kit (Qiagen). The purified products were well separated on a 2% agarose gel, and fragments of 300–1000 bp were excised. The DNA was purified using a QIAquick gel extraction kit, and gel-based size selection and purification was repeated once. Then, 100 μL T1 magnetic streptavidin beads (Invitrogen) were added to each forward and reverse PCR product, and the mixture was incubated for 1 h with gentle rocking at room temperature. The beads were magnetically isolated, washed three times in 500 μL 1× B&W buffer (10 mM Tris pH 7.5, 1 mM EDTA, 2 M NaCl) and once in H₂O and resuspended in 50 μL H₂O. Subsequently, 25 μL of the beads from each of the forward and reverse PCRs were separately mixed with 10 μL Phusion HF buffer (5×), 1.5 μL dNTPs (10 mM), 1 μL forward (20 μM) or reverse MiSeq HBV primer (20 μM), 1 μL forward or reverse MiSeq-pLinker (20 μM) (all MiSeq primers contain an adaptor for Illumina flow cell surface annealing) (Supplementary Table 2), 0.5 μL Phusion Taq (2000 U/mL), and 11 μL H₂O and subjected to PCR (98 °C for 1 min, 35 cycles of 98 °C for 10 s, 65 °C for 40 s, and 72 °C–40 s, followed by 72 °C for 5 min and a hold at 4 °C). The PCR products were magnetically separated from the beads and purified using the QIAquick PCR purification kit (Qiagen). The adaptor-ligated fragments were enriched by 25 cycles of PCR with Illumina primers Index 1 and Index 2, as follows: 98 C° for 1 min, 25 cycles of 98 °C for 30 s, 55 °C for 30 s, and 72 °C for 30 s), and 72 C° for 5 min. Forward and reverse libraries for the same sample were mixed in equimolar ratios and sequenced by 250-bp paired-end sequencing using an Illumina MiSeq. A total of 340 integration libraries were constructed from liver tissue samples of the nine individuals analysed (7 patients with HBV-related HCC and 2 HBsAg-negative subjects as a control) and from the PLC/PRF/5, HepAD38 and Vero cell lines.

**Integration library verification**. To verify our HBV integration sequencing strategy, we constructed 60 libraries using liver DNA from two HBV-uninfected patients and HBV-negative Vero cells. Furthermore, to analyse the saturation of our approach, two separate integration libraries were constructed using identical samples from the two HBV-un-infected patients. No sequences that mapped to viral integration sites were recovered from either the uninfected patients or Vero cells. The HBIS approach has also been verified by analysing HBV integration in PLC/PRF/5 hepatoma cells, which contain multiple integrated HBV DNA fragments and have been extensively investigated using different molecular approaches, including the most sensitive NGS strategies[56,58]. A total of 104 HBV integration

sites were detected in this cell line. Twelve of the 104 sites were found to be HBV integration breakpoints covered by at least three reads (Supplementary Table 3 and Supplementary Data 4). Among these 12 unique integration events, previously described viral integrations in the *TERT* promoter region, in *MVK*, *CCDC57* and *UNC5D* genes and downstream of the *STARD13* gene have been detected[56,58,67]. HBV DNA integrating in the promoter region of the *TERT* gene included a sequence [nucleotide (nt) 1260–1391] of the *ENH1/X* promoter region. In the *MVK* and *CCDC57* genes, the HBV integrants included two distinct sequences of the *S* gene (nt 711–817 and nt 491–456, respectively). In the *UNC5D* gene, the HBV integrant included a sequence of the Core gene (nt 2436–2392) (Supplementary Table 3 and Supplementary Data 4). Moreover, other HBV integration sites were identified in PLC/PRF/5 cells, including in the *Ninein Like* (*NINL*) gene, LOC105375716 ncRNA, the *TBC1* domain family member 8 (TBC1D8) gene, ribosomal protein S23 pseudogene 4 (*RPS23P4*), and ncRNAs LOC105374110 and LOC101929814 (Supplementary Tables 3 and Supplementary Data 4). Viral integrations in the *TERT* promoter and the *CCDC57* gene were confirmed by PCR and Sanger sequencing (Supplementary Fig. 7).

**Validation of HBIS-identified genomic integration sites by PCR and Sanger sequencing**. Genomic DNA isolated as described above was serially diluted and subjected to nested PCR with genome-specific primers and HBV-specific primers (Supplementary Table 2) using HotStart Taq Polymerase (Qiagen) (98 °C for 14 min, 40 cycles of 98 °C for 30 s, 55 °C for 30 s, 2 °C 30 s, and 72 °C for 5 min). The products were isolated by gel electrophoresis and sequenced with the Sanger method. For each PCR amplicon, we used an Applied Biosystems 3500 DNA analyser to perform dye terminator Sanger sequencing.

**RNA extraction and sequencing**. Total RNAs were extracted from frozen tumours or paired adjacent non-tumour samples using TRIzol reagent (Invitrogen, Carlsbad, CA, USA) according to the manufacturer's recommendations. Total RNA quality and integrity were assessed using the Agilent 2100 Bioanalyzer (Santa Clara, CA, USA). Sequencing was performed on the HiSeq 2500 platform using TruSeq Stranded mRNA kit (Illumina).

**Computational analysis: determination of viral integration sites**. To detect HBV integration sites in the human genome, Illumina raw paired-end reads (2 × 250 bp) were initially processed to remove adaptors and low-quality bases with the Trimmomatic program (v.0.36)[68] and the following options: a sliding window of 5 bp, an average base quality score of 16, and a minimum read length of 35 bp. Subsequently, BWA software (version: 0.7.12-r1039)[69] was used to align the clean reads (paired-end and/or unpaired reads) to a custom reference genome including the human GRCh38.p10 reference (GenBank accession: GCA_000001405.25) and the HBV genome genotype D (GenBank accession: NC_003977.2). The Picard tool (version 2.22.0) (http://broadinstitute.github.io/picard) was employed to remove optical duplicate reads from the BAM files; SAMtools software[70] was utilised to extract chimeric reads (SAM flag 2048; supplementary alignment 0×800). Subsequently, the BEDTools utility[25] was used to extract primary and secondary mapping coordinates to properly count the number of integration events occurring in the hybrid genome (the flag -cigar was used in the bamtobed tool). The resulting genomic coordinates, including chromosomal locations related to the integration of HBV DNA into the human genome, were then used to retrieve reads related to each integration event to reconstruct the chimeric sequences using Cap3[71] and cd-hit[72] software. The assembled chimaeras were mapped back to the hybrid genome using the BLAST algorithm[73] with the following options: task=blastn-short, dust=no, soft_masking=false, word_size=7, penalty = -3, reward=2, gapopen=5, gapextend=2. Microhomology was determined by DNA mapping of each chimaera in both the human and HBV genomes.

All the identified putative integration sites were filtered out, and only those chimeric integrations covered by at least 3 reads mapping to the flanking viral sequence at least 32 bp were retained. On average, chimaeras represented 0.03% (range, 0.0045–0.48) of the total number of good-quality reads. All the bioinformatics tools described above have been integrated into an automatic computational pipeline for accurate and efficient detection of viral integration events in the human genome that is available at the GitHub repository (https://github.com/DomeJoyce/HBVIF).

Furthermore, integrations related to putative mitochondrial genes were double-checked by blacklisting known NUMTs reported in ref. [74] and in ref. [75]; then, we aligned these NUMTs against our detected mitochondrial integrations in order to confirm that integrations were properly located in the mitogenome, rather than the nuclear mitochondrial sequences. Finally, to account for the total of mitochondrial integration events reported so far, we collected from this study and other studies[16,60] all HBV integration events that occur in the mitogenome. All genomic regions affected by such events were reported in Supplementary Data 5 and in Supplementary Fig. 8.

**Clonal integration sites and chromosomal enrichment analysis**. For each detected viral integration site, all supporting reads were aligned using the MAFFT aligner[76] (options used:–localpair,–reorder,–maxiterate 1000) to manually inspect the exact position of the breakpoint at the nucleotide level. An integration site was

considered clonally expanded when most of the reads (≥92%) supporting the integration event showed an identical breakpoint flanked by identical sequence motifs. Thus, all identified HBV integration events were classified into two categories: clonal integrations and single integrations. For each chromosome, the chi-square test was used to determine the presence of statistically significant differences ($P$ value threshold≤ 0.05) in the number of integration events occurring between the tumour and non-tumour samples. Moreover, the chi-square test was employed to assess any statistically significant difference in the distribution of HBV integration sites among human chromosomes. The non-parametric combination (NCP) test was applied to assess the probability of HBV integration occurrence into each individual chromosome.

**Monte Carlo simulation for virus integration and hotspots**. Identification of hotspot sites was performed using the BEDTools suite[25]. Monte Carlo simulation was conducted by randomly shuffling the genomic locations of all integration sites (100, 1000 and 10,000 times) using the BEDTools shuffle utility[25]. Subsequently, we compared the observed number of integrations with the median number of integrations in the randomised list. We evaluated enrichment by counting the frequency of observed events equal to or higher than the number of randomised events divided by $n = 100$, 1000, or 10,000.

To determine whether a particular sequence motif of human or viral origin was over-represented in the chimeric sequences, we used the software HOMER (http://homer.ucsd.edu/homer) and the datasets corresponding to the human-human, virus–virus and human–virus chimeric sequences. Specifically, each chimeric dataset was analysed separately, and for statistical analysis, the appropriate genome-specific data (human, HBV and the hybrid human–virus genome) were used as background sequences.

**Subcellular fractionation and mitochondrial isolation**. Sufficient material to perform subcellular fractionation and mitochondrial isolation was only available from tumour tissue specimens obtained from patients 0501, 0504, and 0505. In addition to the tumour tissues from these patients, subcellular fractionation and mitochondrial isolation were performed on liver tissue specimens from five CHB patients with high levels of HBV replication, one HBV-negative patient, and HepG2 and HepAD38 cells following the method previously described in ref. [77]. Briefly, 150 mg of frozen liver tissue from each patient and $8 \times 10^7$ HepG2 or HepAD38 cells were used for mitochondria isolation. The liver tissue specimens were homogenised in cold MitoPrep buffer (0.225 M mannitol, 0.075 M sucrose, 20 mM HEPES pH 7.4) with 1 mM EDTA and 0.5 mM PMSF using a TissueRuptor homogeniser (Qiagen). HepG2 and HepAD38 cells were collected by centrifugation at 2000×$g$ for 2 min at room temperature, washed with PBS 1× pH 7.4 (Lonza), resuspended in MitoPrep buffer (0.225 M mannitol, 0.075 M sucrose, 20 mM HEPES pH 7.4) with 1 mM EDTA and 0.5 mM PMSF and homogenised using a 23-G needle. The homogenates were subjected to the same protocol for nuclear and mitochondrial isolation. Specifically, the tissue and cell homogenates were centrifuged at 800×$g$ for 10 min at 4 °C. The supernatants were transferred into a new tube and placed on ice, and the pellets were re-homogenised using a 27-G needle, resuspended in MitoPrep buffer and centrifuged at 800×$g$ for 5 min at 4 °C. The pellets containing nuclei were stored at −20 °C for subsequent analysis. The two supernatants for each sample were pooled and centrifuged at 800×$g$ for 5 min at 4 °C. The resulting supernatants were transferred to new tubes and centrifuged at 11,000×$g$ for 5 min at 4 °C. The obtained supernatants containing the cytoplasmic fractions were stored at −20 °C for subsequent analysis; the mitochondrial pellets were resuspended in 200 μL of MitoPrep buffer and incubated with 300 U micrococcal nuclease S7 (Thermo Scientific), 10 mM CaCl₂ and 1 μL 10 mg/mL digitonin (Sigma-Aldrich) for 25 min at 27 °C. The reaction was stopped with 5 mM EDTA. The mitoplasts were pelleted at 11,000×$g$ for 4 min at 4 °C and used for DNA extraction, RNA extraction, in vitro RNA import assays, or RNA pull-down assays.

**Plasmids**. The *preS1* and *X* genomic regions of HBV genotype D (ayw) were amplified by PCR, and the resulting 1169-bp and 468-bp fragments (nucleotide positions 2850–837 and 1372–1840 on the HBV map, respectively) were cloned into the pcDNA3.1 directional TOPO vector (Life Technologies), generating pcDNA-preS1 and pcDNA-X. The HBV genomic regions corresponding to the encapsidation signal "epsilon" sequence of the pregenomic RNA (pgRNA) and to stem-loop α (*SLα*) of the post-transcriptional regulatory element (PRE) were also amplified by PCR, and the resulting 287-bp and 680-bp fragments (nucleotide positions 1781–2068 and 670–1350, respectively) were cloned to construct pcDNA-Epsilon and pcDNA-SLα. In addition, the human mitochondrial *COX3* RNA sequence (nt position 9397–9796) was PCR amplified from HepG2 cells, and the obtained 399-bp fragment was inserted into the bidirectional TOPO vector to generate pcDNA-COX3.

**In vitro transcription**. Transcripts were synthesised using T7 RNA Polymerase (Roche Diagnostics) following the manufacturer's instructions. For biotin-labelled RNA synthesis, a Pierce 3' END Deshtiobiotinylation Kit (Thermo Fisher Scientific) was used. RNA was purified using the TRIzol reagent following the manufacturer's instructions.

**In vitro RNA import assay**. In vitro RNA import assays were performed in a 200-μL volume containing 50 μg mitoplasts, 0.225 M mannitol, 0.05 M sucrose, 20 mM HEPES pH 7.4, 25 mM KCl, 5 mM MgCl₂, 5 mM ATP, 15 mM succinate, and 2 mM DTT. After incubation at 30 °C for 5 min, 200 ng RNA was added to the import buffer, and the reactions were incubated at 30 °C for 10 min. Next, 300 U micrococcal nuclease S7 and 10 mM CaCl₂ were added, and the samples were incubated for 30 min at 27 °C. The mixture was transferred into a new tube, and the mitoplasts were sedimented at 11,000×$g$ for 5 min at 4 °C, dissolved in urea-SDS loading buffer (60 mM Tris-HCl pH 6.8, 2% SDS, 5 mM EDTA, 25% glycerol, 0.01% bromophenol blue, 10 μg/mL proteinase K, and saturated urea) and heated at 50 °C for 5 min. The RNA and proteins were resolved by SDS–PAGE. The samples were then analysed using a North2South Chemiluminescent Hybridization and Detection Kit (Thermo Fisher Scientific).

**In vitro RNA pull-down assay**. RNA transcribed from pcDNA-preS1, pcDNA-SLα, pcDNA-X, pcDNA-Epsilon, and pcDNA-COX3 was labelled at the 3' end using a Pierce 3' END Deshtiobiotinylation Kit (Thermo Scientific) following the manufacturer's instructions. Mitoplast pellets from $8 \times 10^7$ HepG2 cells were used for RNA pull-down assays. The pellets were resuspended in 200 μL MitoPrep buffer and incubated with 300 U micrococcal nuclease S7 (Thermo Fisher Scientific) and 10 mM CaCl₂ at 27 °C for 20 min. The protoplasts were then pelleted at 11,000×$g$ for 5 min at 4 °C. The pellets were dissolved in 30 μL MitoPrep buffer and sonicated on ice for 2 s. RNA pull-down was performed using a Pierce Magnetic RNA–Protein Pull-down kit (Thermo Fisher Scientific) following the manufacturer's instructions. Briefly, for each assay, 50 pmol biotinylated RNAs or control RNA was incubated with 50 μL pre-washed streptavidin-agarose beads at 4 °C for 1 h. The RNA-bound beads were incubated with lysates from mitoplasts, and the eluted proteins were detected by western blotting.

**Western blotting**. Cells were washed twice in 1× PBS, pH 7.4, and lysed in Triton X-100 1x cell lysis buffer (Cell Signaling Technology) supplemented with 1 mM PMSF. Mitochondria were lysed directly in 1× SDS loading buffer. Protein lysates (50 μg) as well as the in vitro pull-down eluted proteins were resolved by SDS–PAGE, transferred to PVDF Immobilon-P membranes (Millipore), and incubated for 1 h with 5% Blotting Grade Blocker non-fat milk (Bio-Rad) TBS-T and then overnight with primary antibodies in 5% BSA at 4 °C or for 1–2 h at room temperature. The antibodies used included rabbit anti-PNPase (GTX118737, 1:500, GeneTex), rabbit anti-Mortalin (3593, 1:500, Cell Signaling Technology), anti-β Tubulin mouse (86298, 1:1000, Cell Signaling Technology), mouse anti-Hsp90 (ab13492, 1:500, Abcam). Detection was performed using anti-rabbit or anti-mouse IgG (AP307P, AP308P, 1:3000, 1:1000, Millipore), respectively. The immunoreactive signals were detected using Pierce SuperSignal West Pico Chemiluminescent Substrate (Thermo Fisher Scientific) following the manufacturer's instructions.

**Statistics and reproducibility**. Data are presented as mean ± standard deviation (SD) and categorical variables as absolute frequency and percentage. A non-parametric approach was applied to examine the variable distribution, as verified by the Kolmogorov–Smirnov test. $\chi^2$ or Fisher's exact tests were applied to compare groups of categorical variables. Student's $t$ test was applied for comparison between groups of numerical variables. The NPC test with the Fisher combination function was applied to combine the $P$ values from different comparisons. The statistical analyses were performed using SPSS 22.0 for the Window package. A $P$ value of less than 0.05 was considered to be statistically significant. All statistical tests were two-sided. Real-time PCR quantification experiments, RNA import assay, RNA pull-down analysis and western blot experiments were performed at least two times with similar results.

**Reporting summary**. Further information on research design is available in the Nature Portfolio Reporting Summary linked to this article.

## Data availability

The sequencing data reported in this paper have been deposited in the NCBI SRA database (DNA-seq accession [PRJNA650273]; SRA Study: SRP283983). The authors declare that all other data supporting the findings of this study are available within the article and its Supplementary Information files or are available from the corresponding author on reasonable request. Uncropped scans are provided in Supplementary Figs. 9–13.

## Code availability

Key software or algorithms used in our analysis of sequencing data are listed in Methods. All the bioinformatics tools described in Methods have been integrated into an automatic computational pipeline that is available at the GitHub repository (https://github.com/DomeJoyce/HBVIF).

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

## Acknowledgements

This study was in part supported by a grant awarded by Italian Ministry of Education, University and Research (MIUR), PRIN 2017 (2017MPCWPY_003) to G. Raimondo.

## Author contributions

T.P. and G. Raimondo conceived the study. T.P., D.G., D.L., D.D., C.M., G. Raffa, G.C., R.A.C. and O.R. contributed to pipelines and/or data management. D.G., R.A.C. and O.R. analysed the sequencing data. D.G., O.R. and T.P. analysed and interpreted the data. D.G. and A.A. performed bioinformatics and statistical analysis. C.S. and G. Raimondo analysed the clinical data. C.M., V.C., G. Raffa, F.C.T. and D.D. performed sequencing. D.L., C.M., G. Raffa and V.C. performed laboratory experiments. G.N. provided tumour specimens. T.P. wrote the manuscript with assistance from all authors. T.P. supervised the project.

## Competing interests

The authors declare no competing interests.
