## [Peer Review File · Communications Biology]

Reviewers' comments:

Reviewer #1 (Remarks to the Author):

HBV infection is a major risk factor for HCC development. This study applied a new high-throughput HBV integration sequencing approach to identify HBV integration sites and the number of integration clones. The authors have provided strong evidence showing that HBV sequences are integrated into mitochondrial DNA and the sites of integration are mostly in OXPHOS genes in tumor and D-loop in non-tumor. They also show that the integration is viral RNA mediated and is dependent on PNPASE-regulated mitochondrial RNA import. Their findings provide a possible new mechanism by which HBV contributes to HCC development.

There are a few minor mistakes and questions to be addresses.

1. Supplemental Figure 1 does not seem to support the conclusion in Paragraph 2, Page 6.
2. Supplemental Figure 5 is mislabeled.
3. There is a grammar error in Line 23, Page 9.
4. Is it possible to get information from the sequencing data about the sizes of the integrated HBV sequences in mtDNA and the frequency of each integrated HBV gene?
5. It appears that most integrations in mtDNA are fairly small HBV sequences. Are there much longer ones, those as big as the imported viral RNAs?
6. If integration in OXPHOS genes is selected against in non-tumor cells, shouldn't integration in D-loop be as frequent as in OXPHOS genes in tumor cells? How come the sites of integration are mostly in OXPHOS genes in tumor cells?
7. It would help readers understand better if the authors could add some discussion about why disruption of OXPHOS genes is better tolerated in cancer cells. Could the integration provide some metabolic advantages for cancer development?
8. Some explanation on Tet removal in the text (Paragraph 3, Page 8) will also help readers not familiar with HBV research.

Reviewer #2 (Remarks to the Author):

This study detected 3,339 HBV integration sites in 7 HCC patients by using their high-throughput sequencing technology. Many studies reported the similar results with this study, but they focused on HBV integration into mtDNA.

- 1) Furuta et al. (ref. 44, Oncotarget 2018) reported HBV integration into mtDNA in the mice HBV infection model and this suggested a possibility that mtDNA is a preferential or target genome for HBV integration in the early phase of HBV infection. They analyzed clonality of each integration and how about the clonality of mtDNA integration?
- 2) What is the reason that they selected chimeric reads covered by at least 3 reads mapping to the flanking viral sequence with at least 32 bp?
- 3) It is not clear how they define the clonal integration sites. Probably they are required to use more statistical method.
- 4) They just analyzed and discussed only 20 HBV integration sites in mtDNA. The sample number is too small to discuss, and they should include the data of the previous reports and database.

5) How should they interpret the integration to D-loop of mtDNA?

Reviewer #3 (Remarks to the Author):

The manuscript entitled: "Mitochondrial DNA is a frequent target of HBV integration" is a fascinating and potentially important proof-of-concept paper. Using a high-throughput HBV integration sequencing (HBIS) method and a new bioinformatic pipeline, Giosa D et al. tried to comprehensively profile the characteristics of HBV integration events in tumor and adjacent nontumor liver tissues from HBV-related HCC patients, which is original and valuable. The observations that mtDNA derived from human liver tissues is a preferred site for HBV integration and that HBV RNA could be involved in the integration process are of great interest. This manuscript is well written and requires some moderate revisions to be accepted for publication.

Major concerns:

1. Study design limitations:

Although it is a proof-of-concept study, the patient cohort (n=7) is too small to achieve a solid conclusion, especially the finding that mitochondrial DNA (mtDNA) was a recurrent HBV integration target.

2. Methodological and analytical limitations:

1) Nuclear DNA contamination: Although the authors attempted to demonstrate that the HBV integrants in the mitochondria detected by HBIS and RNASeq originated instead from potential NUMTs (ref is needed in manuscript), it is hard to avoid Nuclear DNA contamination on HBV integration sites detection in mtDNA. High homology between mtDNA and Nuclear Mitochondrial sequences (NUMTs) causes problems during alignment, such as abnormally high mtDNA coverage due to misalignment of the nuclear reads to mtDNA, which in return can confound the analysis of HBV integration breakpoints in mtDNA. To address this issue, more robust bioinformatic methods are needed. For example, the authors can try methods such as blacklisting the known NUMT regions provided in Li et al. (Nucleic Acids Res. 2012, 40, e137) or Dayama et al. (Nucleic Acids Res. 2014, 42(20), 12640-12649). More strict alignment strategies coupled with high mapping quality thresholds during filtering can also be used. If this method proves to be inefficient, it is possible to validate the called variants through other methods, such as using BLAST to check if the variant occurs elsewhere in the nuclear genome or filtering them based on variant allele frequency, latter previously demonstrated by Yuan et al (Nat Genet 2020. 52, 342-352.).

2) Unreasonable comparisons: In result 2, it is unreasonable to conclude that clonally expanded viral integrations favored the gene body in tumor tissues by comparing proportions of clonally expanded and single viral integrations in gene body in tumor tissues, so did single integrations. To find out whether clonally expanded viral integrations preferred the gene body in tumor tissues, it makes sense to compare the proportion of clonally expanded integrations in gene body versus that in intergenic regions.

Minor concerns:

1. In Fig. 2., the histogram of non-tumor tissue did not have a bar value.

2. To better understand the results, I recommend showing two different colored lines in Fig. 3 to indicate HBV integration in tumor and non-tumor tissues.

3. "Supplementary Fig 5" was mislabeled as "Supplementary Fig 4" in the figure legend of "Supplementary Fig 5".

4. The enrollment information of 5 chronic hepatitis B (CHB) and 8 HBV-negative patients used to investigate the HBV integration mechanism (Result 5) was not mentioned in the "Methods" section.

5. "Supplementary Table 9" mentioned on page 8 (Result 4) was not shown in the manuscript as well

as the supplementary materials.

6. The language of the manuscript needs to be polished to avoid unintentional mistakes (e.g., the perfect sense of "show" is "shown", not "showed", as in the subtitle of Supplementary table 1).

7. The amounts of total HBV DNA in tumor and non-tumor tissues seem to be statistically different (468.2 ± 792 versus $2,269 \pm 5,622$, Table 2).

Point-by-Point Response to the Reviewers:

We thank the reviewers for their positive evaluation of our work and for the suggestions, which were very helpful in improving the quality and strength of our study. The individual comments are shown in bold.

Reviewer 1:

HBV infection is a major risk factor for HCC development. This study applied a new high-throughput HBV integration sequencing approach to identify HBV integration sites and the number of integration clones. The authors have provided strong evidence showing that HBV sequences are integrated into mitochondrial DNA and the sites of integration are mostly in OXPHOS genes in tumor and D-loop in non-tumor. They also show that the integration is viral RNA mediated and is dependent on PNPASE-regulated mitochondrial RNA import. Their findings provide a possible new mechanism by which HBV contributes to HCC development.

There are a few minor mistakes and questions to be addresses.

We appreciate the favorable comments of the reviewer on our work, and we thank the reviewer for the constructive criticisms, which were very helpful to improve the quality of our study.

- 1 Supplemental Figure 1 does not seem to support the conclusion in Paragraph 2, Page 6.**

We thank the reviewer for this remark. We apologise for this inaccuracy. In the revised manuscript, we have included the correct Supplemental Figure 1.

- 2 Supplemental Figure 5 is mislabeled**

We also apologise for this inaccuracy. We have now correctly labelled Supplemental Figure 5

- 3 There is a grammar error in Line 23, Page 9**

We thank you for your careful review. In the revised manuscript, we have amended the error in Line 32, Page 9.

- 4 Is it possible to get information from the sequencing data about the sizes of the integrated HBV sequences in mtDNA and the frequency of each integrated HBV gene?**

We thank the reviewer for this request, which allows us to provide clearer information concerning sequencing data about HBV integration in mtDNA. We are now providing a revised version of Supplementary Data 4, in which we have better detailed the

required information. In particular, in Supplementary Data 4, sequencing data concerning the sizes of the integrated HBV sequences and the frequency of each integrated HBV gene are reported.

5 It appears that most integrations in mtDNA are fairly small HBV sequences. Are there much longer ones, those as big as the imported viral RNAs?

As noticed by the reviewer, by HBIS we could only detect “fairly small” HBV sequences ranging between 42bp and 187bp, and we did not find any integration as big as the imported viral RNAs. We also thank the reviewer for giving us the opportunity to better clarify the main aspects of our HBV-enrichment sequencing approach that are reported in the Supplementary Information. In particular, we obtained cellular DNA fragments ranging from 100bp to 1,000bp after shearing DNA by a SONOPULS ultrasonic homogeniser (Bandelin). These fragments were purified, end blunted, “A” tailed and “Linker” ligated. Then, in order to recover the HBV integration sites, the products of ligation were amplified by PCR and the use of HBV specific primers and primers specific to the “Linker”. The principle of the applied PCR technique does not allow simultaneous identification of both left and right virus-cell DNA junctions of the integrated HBV DNA. Therefore - as for other PCR approaches (e.g. InvPCR or ALU PCR) - our method is unable to detect the full integrated HBV sequence. Indeed, profiling left and right virus-cell DNA junctions of the integrated HBV DNA in separate reactions has a low probability of detecting the same integration event, as few copies of each integration event exist in the thousands (~5,000) of copies of mtDNA present in each infected cell. However, by cloning and Sanger sequencing we succeeded in identifying both left and right virus-cell mtDNA junctions in the tumour tissue from patient 0504. In particular, the cloning approach led us to identify the entire HBV sequence integrated in the mitochondrial *ND5* gene (Supplementary Fig. 5). The length of this integrated HBV sequence was 363bp and included two gamma interferon activation site (GAS) elements and the *ENH1* region. We included the details of our findings in the revised manuscript (Page 7, Line 11 and Page 8, Lines 1 - 4) and in the Figure legend of Supplementary Fig. 5.

6 If integration in OXPHOS genes is selected against in non-tumor cells, shouldn't integration in D-loop be as frequent as in OXPHOS genes in tumor cells? How come the sites of integration are mostly in OXPHOS genes in tumor cells?

We thank the reviewer for giving us the opportunity to better detail and clarify our results. We found that most of the integration sites detected in mtDNA from tumour tissues were located within the mitochondrial genes of OXPHOS system and that mtDNA from non-tumour tissues showed no integration in these genes (11/17 tumour tissues versus 0/3 non-tumour tissue; $P=0.073$, *Fisher's exact test*). Thus – as reported by the reviewer- integration in OXPHOS genes appears to be selected against in non-tumour cells. Concerning integration in the *D-loop* region, if the number of HBV integrations in each single OXPHOS gene and not in the entire OXPHOS system is taken into consideration, it results that integration in the *D-loop* region is as

frequent as that in any other gene of the OXPHOS system in tumour cells. Thus, we agree with the concern raised by the reviewer and we have modified the sentences on Page 11, Lines 16-20 in the previous version of the manuscript (now, Page 11, Lines 18-23). In the revised manuscript, we also discussed about HBV integration in the D-loop region on Page 12, Lines 10-18.

Concerning the second question, “How come the sites of integration are mostly in OXPHOS genes in tumor cells?” It is conceivable to hypothesise that liver cancer cells can tolerate OXPHOS genes disruption induced by HBV integration. However, it remains to be established whether these mitochondrial alterations are simple passenger or rather driver mutations, which may favor cell transformation, cancer initiation and/or progression. Important available evidence strongly suggests that mitochondrial DNA aberrations may have functional oncogenic impacts in the initiation and clonal evolution of the specific cancer types. In the revised manuscript we have discussed the possible role of disruption of OXPHOS genes by HBV integration and cancer development (Page 11, Lines 29-34 and Page 12, Lines 1-9).

7 It would help readers understand better if the authors could add some discussion about why disruption of OXPHOS genes is better tolerated in cancer cells. Could the integration provide some metabolic advantages for cancer development?

As requested by the reviewer, and previously addressed in point 6, in the revised manuscript we have discussed the possible role of disruption of OXPHOS genes by HBV integration and cancer development (Page 11, Lines 29-34 and Page 12, Lines 1-9).

8 Some explanation on Tet removal in the text (Paragraph 3, Page 8) will also help readers not familiar with HBV research.

As requested, we have included more details concerning Tet removal in the Results section (Page 8, Lines 23-34) of the revised manuscript.

Reviewer 2:

This study detected 3,339 HBV integration sites in 7 HCC patients by using their high-throughput sequencing technology. Many studies reported the similar results with this study, but they focused on HBV integration into mtDNA

We thank the reviewer for the helpful suggestions. As detailed in the point-by-point response we have addressed all the issues raised by the reviewer. The reviewer's comments are shown in bold.

1 Furuta et al. (ref. 44, Oncotarget 2018) reported HBV integration into mtDNA in the mice HBV infection model and this suggested a possibility that mtDNA is a preferential or target genome for HBV integration in the early phase of HBV

infection. They analyzed clonality of each integration and how about the clonality of mtDNA integration?

In the revised manuscript Furuta et al, Oncotarget 2018 is numbered as ref. 60. We thank the reviewer for the remark and for giving us the opportunity to specify that we also analysed clonality of HBV integration in mtDNA. Indeed, the HBIS method that enables the enumeration of identical viral integration sites and allows for the identification of clonally expanded integrations was also applied to detect HBV integration in mtDNA. In the revised manuscript we have specified that all integrations detected in mtDNA from tumour and non-tumour tissue samples were clonally expanded (Page 7, Lines 17-19).

2 What is the reason that they selected chimeric reads covered by at least 3 reads mapping to the flanking viral sequence with at least 32 bp?

We selected chimeric reads covered by at least 3 reads with a flanking viral sequence of at least 32 bp in order to impose a stringent condition for their selection. The choice of flanking sequences, with at least 32 bp in length, was made by considering the maximum length of the HBV-specific primer sequences (up to 24 bp, so flanking viral sequences need to have a minimum of an additional 8 bp) designed for the HBIS workflow. Moreover, breakpoints, predicted by 3 or more supporting read-pairs, have been reported in several important papers (Fujimoto A et al, Nature Genetics 2012; Sung W-K et al Nature Genetics, 2012; Zhao L-H et al Nature Communication, 2016).

3 It is not clear how they define the clonal integration sites. Probably they are required to use more statistical method.

As reported in Results, we define clonally expanded integrations as “identical integration sites with distinct fragmentation ends, deriving from the clonal expansion of a single integration event” (Page 4, Lines 18-24). Furthermore, in Methods, “Clonal integration sites and chromosomal enrichment analysis” paragraph we detail the bioinformatic and statistical approaches we used to identify clonal integration sites. Basically, the integration event was considered clonal if, for that specific integration site, at least 92% of the total mapped reads at the viral/human breakpoint demonstrated identical flanking sequences in both the viral and human sides.

4 They just analyzed and discussed only 20 HBV integration sites in mtDNA. The sample number is too small to discuss, and they should include the data of the previous reports and database.

We understand the reviewer’s concern and we have discussed the data from previous reports and database in the revised manuscript (Page 13, Lines 6-21).

Moreover, a comparison among data of the previous reports and database has been performed (Page 17, Line 12-15) and summarized in the Supplementary Data 9 and in the Supplementary Fig. 8.

5 How should they interpret the integration to D-loop of mtDNA?

We detected HBV integration in the mitochondrial *D-loop* region both in tumour and non-tumour liver tissues. This led us to suppose that viral integration in this region is an event that may precede hepatocellular transformation and which might go through a positive selection during hepatocarcinogenesis. In the revised manuscript we have discussed this aspect and added references (Page 12, Lines 10-18).

Reviewer 3:

The manuscript entitled: "Mitochondrial DNA is a frequent target of HBV integration" is a fascinating and potentially important proof-of-concept paper. Using a high-throughput HBV integration sequencing (HBIS) method and a new bioinformatic pipeline, Giosa D et al. tried to comprehensively profile the characteristics of HBV integration events in tumor and adjacent nontumor liver tissues from HBV-related HCC patients, which is original and valuable. The observations that mtDNA derived from human liver tissues is a preferred site for HBV integration and that HBV RNA could be involved in the integration process are of great interest. This manuscript is well written and requires some moderate revisions to be accepted for publication.

We thank the reviewer for the positive evaluation of our work and for the helpful suggestions to improve the quality of our study. We have performed extensive additional bioinformatic analysis to address the issue raised in the review. The obtained data have significantly strengthened our study by corroborating the finding that mitochondrial DNA is a target of HBV integration. The reviewer's comments are shown in bold.

Major concerns:

1. Study design limitations:

Although it is a proof-of-concept study, the patient cohort (n=7) is too small to achieve a solid conclusion, especially the finding that mitochondrial DNA (mtDNA) was a recurrent HBV integration target.

We respectfully note that the main aim of our study was the development of a new high-throughput HBV integration sequencing (HBIS) method coupled to a new bioinformatic pipeline for the detection of HBV integration. To this aim we modified an integration sequencing method described to study HIV integration (Cohn et al., Cell 2015). As reported in Methods, to verify the sensitivity our new HBV integration sequencing and bioinformatic strategies we analysed HBV integration in PLC/PRF/5 hepatoma cells, which contain multiple integrated HBV DNA fragments and have been extensively investigated using different molecular approaches, including the most sensitive NGS strategies (Watanabe, Y. et al Genome Res, 2015; Ishii, T. et al. Genes, 2020). We detected a total of 104 HBV integration sites in this cell line. Twelve of the 104 sites were found to be HBV integration breakpoints covered by at least 3 reads. Among these 12 unique integration events, previously described viral integrations in the *TERT* promoter region, in *MVK*, *CCDC57*, and *UNC5D* genes and downstream of the

STARD13 gene have been detected (Graef, E., *Oncogene*, 1994; Watanabe, Y. et al *Genome Res*, 2015; Ishii, T. et al. *Genes*, 2020). Furthermore, to verify the specificity of the HBIS approach we analysed liver tissues from 2 HBV-uninfected patients and HBV-negative Vero cells. No sequences that mapped to viral integration sites were recovered from either the uninfected patients or Vero cells. Then we applied the HBIS method to study HBV integration in tumour and non-tumour tissue samples from 7 patients with HBV-related HCC and in HepAD38 cells that supports tetracycline-off inducible HBV replication. The HBIS allowed the identification of clonal and single integration sites and turned out to be a very sensitive approach. Indeed, we detected a very high number of HBV integration sites (3,339) in the 7 studied patients. Furthermore, the HBIS led us to detect HBV integration into mitochondrial DNA (mtDNA) from both tumour and non-tumour tissue specimens and from HBV-producing HepAD38 cells for the first time. However, we agree with the reviewer that to define HBV integration into mtDNA as a “recurrent target” is perhaps inappropriate, considering the limited number of studied patients. Therefore, the term “recurrent” was deleted when mentioned in the manuscript in reference to HBV integration into mtDNA, and the term “frequent” was deleted from the title.

2. Methodological and analytical limitations:

1) Nuclear DNA contamination: Although the authors attempted to demonstrate that the HBV integrants in the mitochondria detected by HBIS and RNASeq originated instead from potential NUMTs (ref is needed in manuscript), it is hard to avoid Nuclear DNA contamination on HBV integration sites detection in mtDNA. High homology between mtDNA and Nuclear Mitochondrial sequences (NUMTs) causes problems during alignment, such as abnormally high mtDNA coverage due to misalignment of the nuclear reads to mtDNA, which in return can confound the analysis of HBV integration breakpoints in mtDNA. To address this issue, more robust bioinformatic methods are needed. For example, the authors can try methods such as blacklisting the known NUMT regions provided in Li et al. (*Nucleic Acids Res.* 2012, 40, e137) or Dayama et al. (*Nucleic Acids Res.* 2014, 42(20), 12640-12649). More strict alignment strategies coupled with high mapping quality thresholds during filtering can also be used. If this method proves to be inefficient, it is possible to validate the called variants through other methods, such as using BLAST to check if the variant occurs elsewhere in the nuclear genome or filtering them based on variant allele frequency, latter previously demonstrated by Yuan et al (*Nat Genet* 2020. 52, 342-352.).

As suggested by the reviewer, we blacklisted the known NUMTs reported in Li et al (*Nucleic Acids Res.* 2012, 40, e137) and Dayama et al (*Nucleic Acids Res.* 2014, 42(20), 12640-12649), and aligned them against our detected mitochondrial integrations (Page 17, Lines 9-12). This analysis showed that our HBV integrations are located in sequences belonging to mitochondrial DNA. In relation to the reviewer’s point with regards to the use of other validation methods, we can confirm that BLAST search of the reconstructed chimeras against the hybrid genome had previously been adopted by the original pipeline to double-check the correctness of the location of each viral integration.

2) Unreasonable comparisons: In result 2, it is unreasonable to conclude that clonally expanded viral integrations favored the gene body in tumor tissues by comparing proportions of clonally expanded and single viral integrations in gene body in tumor tissues, so did single integrations. To find out whether clonally expanded viral integrations preferred the gene body in tumor tissues, it makes sense to compare the proportion of clonally expanded integrations in gene body versus that in intergenic regions.

We are grateful to the reviewer for this remark. We have now compared the proportion of clonally expanded integrations in gene body versus that in intergenic regions in tumour and non-tumour tissues. The revised manuscript has been modified accordingly (page 5, lines 20-21).

Minor concerns:

1. In Fig. 2., the histogram of non-tumor tissue did not have a bar value.

Thank you for your point. In new Fig.2, we included the bar value in the histogram of non-tumour tissue.

2. To better understand the results, I recommend showing two different colored lines in Fig. 3 to indicate HBV integration in tumor and non-tumor tissues.

Following the recommendation of the reviewer, we modified Fig. 3 by using two different coloured lines to indicate HBV integration in tumour (red lines) and non-tumour tissues (blue lines).

3. "Supplementary Fig 5" was mislabeled as "Supplementary Fig 4" in the figure legend of "Supplementary Fig 5".

We apologise for this inaccuracy. The Supplementary Figure 5 is now correctly labelled.

4. The enrollment information of 5 chronic hepatitis B (CHB) and 8 HBV-negative patients used to investigate the HBV integration mechanism (Result 5) was not mentioned in the "Methods" section.

We thank the reviewer for this point. The Methods section has now been updated to include reference to the enrollment information for both patient groups (Page 9, Lines 13-15, and Page 14, Lines 21-32).

5. "Supplementary Table 9" mentioned on page 8 (Result 4) was not shown in the manuscript as well as the supplementary materials.

We thank the reviewer for their observation. The reference to Supplementary Table 9 on Page 8 (now Page 9, Line 3) was an error. This should have been a reference to Supplementary Data 6. In the new version of the manuscript this has been corrected.

With regards to the Supplementary Materials, this information was previously included in a file entitled Supplementary Information, and this is also the case in the new version.

6. The language of the manuscript needs to be polished to avoid unintentional mistakes (e.g., the perfect sense of "show" is "shown", not "showed", as in the subtitle of Supplementary table 1).

Thank you for your careful review and attention. The amends have been incorporated in the revised manuscript.

7. The amounts of total HBV DNA in tumor and non-tumor tissues seem to be statistically different (468.2 ± 792 versus $2,269 \pm 5,622$, Table 2).

We thank the reviewer for this observation. We repeated the statistical Student's *t*-test, and the obtained results confirmed our initial finding that the amounts of total HBV DNA in tumour and non-tumour tissues are not statistically different. These results are likely due to the limited number of samples analysed.

REVIEWERS' COMMENTS:

Reviewer #1 (Remarks to the Author):

The authors have addressed all my concerns.

Reviewer #2 (Remarks to the Author):

I am satisfied with their proper address.

Reviewer #3 (Remarks to the Author):

The authors have carefully revised their manuscript, which have successfully addressed all my concerns and improved the quality and strength of their study. This manuscript is well written and achieves the requirements for publication.